**Land-sea coupling of early Pleistocene glacial cycles in the southern North Sea exhibit**
**dominant Northern Hemisphere forcing**
Timme H. Donders[1,2], Niels A.G.M. van Helmond[3], Roel Verreussel[2], Dirk Munsterman[4],
Johan Ten Veen[4], Robert P. Speijer[5], Johan W.H. Weijers[3]*, Francesca Sangiorgi[3], Francien
Peterse[3], Gert-Jan Reichart[3,6], Jaap S. Sinninghe Damsté[3,6], Lucas Lourens[3], Gesa Kuhlmann[7]
and Henk Brinkhuis[3,6]
[1] Department of Physical Geography, Fac. of Geosciences, Utrecht University,
Heidelberglaan 2, 3584CD, Utrecht, The Netherlands.
[2] TNO - Applied Geosciences, Netherlands Organisation of Applied Scientific Research
Princetonlaan 6, 3584 CB Utrecht, The Netherlands.
[3] Department of Earth Sciences, Fac. of Geosciences, Utrecht University, Heidelberglaan 2,
3584CS, Utrecht, The Netherlands.
[4] TNO - Geological Survey of the Netherlands, Netherlands Organisation of Applied
Scientific Research, Princetonlaan 6, 3584 CB Utrecht, The Netherlands.
[5] Department of Earth and Environmental Sciences, KU Leuven, 3001 Heverlee, Belgium
[6] NIOZ Royal Netherlands Institute for Sea Research, P.O. Box 59, 1790 AB, Den Burg,
Texel, The Netherlands
[7] BGR - Federal Institute for Geosciences and Natural Resources, Geozentrum Hannover
Stilleweg 2, D-30655 Hannover
* Currently at: Shell Global Solutions International B.V., Grasweg 31, 1031 HW, Amsterdam,
The Netherlands
Correspondence to: t.h.donders@uu.nl

**Abstract**

We assess the disputed phase relations between forcing and climatic response in the early Pleistocene with a spliced Gelasian (~2.6 – 1.8 Ma) multi-proxy record from the southern North Sea basin. The cored sections couple climate evolution on both land and sea during the intensification of Northern Hemisphere Glaciations (NHG) in NW Europe, providing the first well-constrained stratigraphic sequence of the classic terrestrial Praetiglian Stage. Terrestrial signals were derived from the Eridanos paleoriver, a major fluvial system that contributed a large amount of freshwater to the northeast Atlantic. Due to its latitudinal position, the Eridanos catchment was likely affected by early Pleistocene NHG, leading to intermittent shutdown and reactivation of river flow and sediment transport. Here we apply organic geochemistry, palynology, carbonate isotope geochemistry, and seismostratigraphy to document both vegetation changes in the Eridanos catchment and regional surface water conditions and relate them to early Pleistocene glacial-interglacial cycles and relative sea-level changes. Paleomagnetic and palynological data provide a solid integrated timeframe that ties the obliquity cycles, expressed in the borehole geophysical logs, to Marine Isotope Stages (MIS) 103 to 92, independently confirmed by a local benthic oxygen isotope record. Marine and terrestrial palynological and organic geochemical records provide high resolution reconstructions of relative Terrestrial and Sea Surface Temperature (TT and SST), vegetation, relative sea level, and coastal influence.

During the prominent cold stages MIS 98 and 96, as well as MIS 94 the record indicates increased non-arboreal vegetation, and low SST and TT, and low relative sea level. During the warm stages MIS 99, 97 and 95 we infer increased stratification of the water column together with higher % arboreal vegetation, high SST and relative sea-level maxima. The early Pleistocene distinct warm-cold alterations are synchronous between land and sea, but lead the relative sea-level change by 3-8 thousand years. The record provides evidence for a

dominantly NH driven cooling that leads the glacial build up and varies on obliquity
timescale. Southward migration of Arctic surface water masses during glacials, indicated by
cool-water dinoflagellate cyst assemblages, is furthermore relevant for the discussion on the
relation between the intensity of the Atlantic meridional overturning circulation and ice sheet
growth.

**Keywords**: Glacial-interglacial climate, palynology; organic geochemistry; obliquity, land-
sea correlation, Eridanos delta, southern North Sea

## 1 Introduction

The build-up of extensive Northern Hemisphere (NH) land ice started around 3.6 Ma ago (Ruddiman et al. 1986; Mudelsee and Raymo, 2005; Ravelo et al., 2004; Ravelo, 2010), with stepwise intensifications between 2.7 and 2.54 Ma ago (e.g., Shackleton and Hall, 1984; Raymo et al., 1989; Haug et al., 2005; Lisiecki and Raymo, 2005; Sosdian and Rosenthal, 2009). In the North Atlantic region the first large-scale early Pleistocene glaciations, Marine Isotope Stages (MISs) 100 - 96, are marked by e.g. appearance of ice-rafted debris and southward shift of the Arctic front (see overviews in Naafs et al., 2013; Hennissen et al., 2015). On land, the glaciations led to faunal turnover (e.g. Lister, 2004; Meloro et al., 2008) and widespread vegetation changes (e.g. Zagwijn, 1992; Hooghiemstra and Ran, 1994; Svenning, 2003; Brigham-Grette et al., 2013). Many hypotheses have been put forward to explain the initiation of these NH glaciations around the Plio-Pleistocene transition interval. Causes include tectonics (Keigwin, 1982, Raymo, 1994; Haug and Tiedemann, 1998; Knies et al, 2004; Poore et al., 2006), orbital forcing dominated by obliquity-paced variability (Hays et al., 1976; Maslin et al., 1998; Raymo et al., 2006) and atmospheric $CO_2$ concentration decline (Pagani et al., 2010; Seki et al., 2010; Bartoli et al., 2011) driven by e.g. changes in ocean stratification that affected the biological pump (Haug et al., 1999). Changes were amplified by NH albedo changes (Lawrence et al., 2010), evaporation feedbacks (Haug et al., 2005), and possibly tropical atmospheric circulation change and breakdown of a permanent El Niño (Ravelo et al., 2004; Brierley and Fedorov, 2010; Etourneau et al., 2010).

Key aspects in this discussion are the phase relations between temperature change on land, in the surface and deep ocean, and ice sheet accretion (expressed through global eustatic sea-level lowering) in both Northern and Southern Hemispheres. According to Raymo et al.

(2006), early Pleistocene obliquity forcing dominated global sea level and $\delta^{18}O_{benthic}$, because
precession-paced changes in the Greenland and Antarctic ice sheets cancelled each other out.
In this view, climate records independent of sea-level variations should display significant
variations on precession timescale. Recent tests of this hypothesis indicate that early
Pleistocene precession signals are prominent in both Laurentide ice sheet meltwater pulses
and iceberg-rafted debris of the East Antarctic ice sheet, and decoupled from marine $\delta^{18}O$
(Patterson et al., 2014; Shakun et al., 2016). Alternatively, variations in the total integrated
summer energy, which is obliquity controlled, might be responsible for the dominant
obliquity pacing of the early Pleistocene (Huybers, 2011; Tzedakis et al., 2017). The
dominance of the obliquity component has been attributed to feedbacks between high-latitude
insolation, albedo (sea-ice and vegetation) and ocean heat flux (Koenig et al., 2011; Tabor et
al., 2014). Sosdian and Rosenthal (2009) suggested that temperature variations, based on
benthic foraminifer magnesium/calcium (Mg/Ca) ratios from the North Atlantic, explain a
substantial portion of the global variation in the $\delta^{18}O_{benthic}$ signal. Early Pleistocene North
Atlantic climate responses were closely phased with $\delta^{18}O_{benthic}$ changes, evidenced by
dominant 41-kyr variability in North American biomarker dust fluxes at IODP Site U1313
(Naafs et al., 2012), suggesting a strong common NH high latitude imprint on North Atlantic
climate signals (Lawrence et al., 2010). Following this reasoning, glacial build-up should be
in phase with decreases in NH sea surface temperatures (SST) and terrestrial temperatures
(TT).

To explicitly test this hypothesis we perform a high-resolution multiproxy terrestrial and
marine palynological, organic geochemical, and stable isotope study on a marginal marine
sediment sequence from the southern North Sea (SNS) during the early Pleistocene "41 kyr-
world". We investigate the leads and lags of regional marine vs. terrestrial climatic cooling
during MIS 102-92, and assess the local sea-level response relative to global patterns from the
$\delta^{18}O_{benthic}$ stack of Lisiecki and Raymo (2005; LR04). In a dominantly, NH obliquity driven
scenario, we expect the marine and terrestrial temperature proxies to be in phase on obliquity
timescales with a short (less than 10 kyr) lead on sea-level variations. In addition, the record
can better constrain the signature and timing of the regional continental Praetiglian stage (Van
der Vlerk and Florschütz, 1953; Zagwijn, 1960) that is still widely used, although its
stratigraphic position and original description are not well defined (Donders et al., 2007;
Kemna and Westerhoff, 2007).

## 2 Geological setting

During the Neogene the epicontinental North Sea Basin was confined by landmasses except
towards the northwest, where it opened into the Atlantic domain (Fig. 1) (Bijlsma, 1981;
Ziegler, 1990). Water depths in the central part were approximately between 100 to 300 m as
deduced from seismic geometry (Huuse et al., 2001; Overeem et al., 2001). In contrast, the
recent North Sea has an average depth between 20-50 m in the south that deepens only
towards the shelf edge towards 200 m in the north-west (e.g., Caston, 1979). From the
present-day Baltic region a formidable river system, known as the Eridanos paleoriver,
developed which built up the Southern North Sea delta across southern Scandinavia (Sørensen
et al., 1997; Michelsen et al., 1998; Huuse et al., 2001; Overeem et al., 2001).

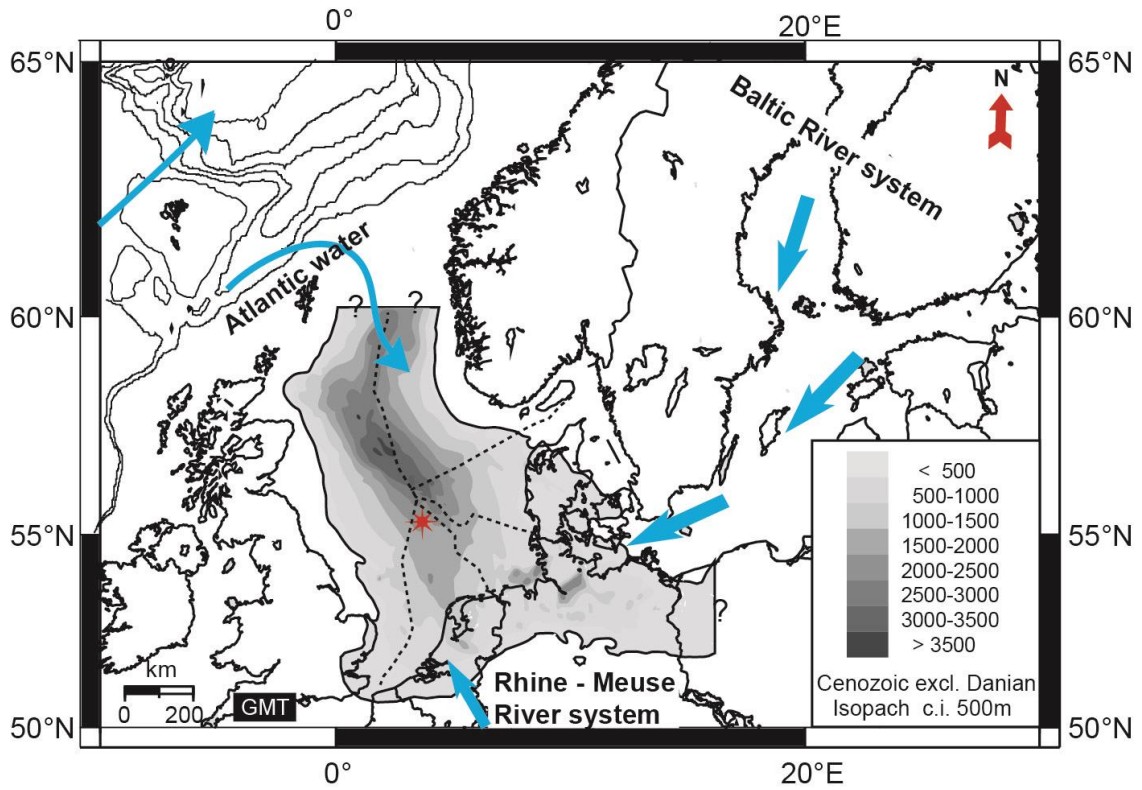


*Figure 1: Geographical map of the present day North Sea region with the superimposed*

*thickness of Cenozoic sediment infill after Ziegler (1990) and the offshore sectors (dashed*

*lines). The reconstructed different water sources (see Gibbard and Lewin, 2016) that*

*influenced the Pliocene and early Pleistocene North Sea hydrography ,including the*

*freshwater supply of the Baltic river system, the Rhine-Meuse river system and Atlantic*

*surface waters are indicated with blue arrows. The location of both boreholes A15-3 (UTM X*

*552567.1, Y 6128751.6) and A15-4 (UTM X 557894.4, Y 6117753.5) is marked by an asterisk,*

*see Fig. S1 for details.*

This delta was characterized by an extensive distributary system that supplied large amounts

of freshwater and sediment to the shelf sea during the Neogene and early Pleistocene

(Overeem et al., 2001), resulting in a sediment infill of ~1500 m in the central North Sea

Basin (Fig. 1). This system was fed by rainfall as well as by melt-water originating from

Scandinavian glaciers (Kuhlmann et al., 2004), principally from the Baltic Shield in the east

with some contribution from the south (Fig. 1) (Bijlsma, 1981; Kuhlmann, 2004). The
sedimentation rates reached up to 84 cm/kyr at the studied locations (Fig. 2) (Kuhlmann et al.,
2006b). Today, the continental river runoff contributes only 0.5 % of the water budget in the
North Sea (Zöllmer and Irion, 1996) resulting in sedimentation rates ranging between 0.4 to
1.9 cm/kyr in the Norwegian Channel, and 0.5 - 1 cm/kyr in the southern part of the North
Sea (de Haas et al., 1997).

**3 Material, core description and age model**
Recent exploration efforts in the SNS led to the successful recovery of cored sedimentary
successions of marine isotope stages (MIS) 102-92 and continuous paleomagnetic logs (Fig.
2) (Kuhlman et al, 2006ab). For quantitative palynological and geochemical analyses, discrete
sediment samples were taken from two exploration wells A15-3 and A15-4 located in the
northernmost part of the Dutch offshore sector in the SNS at the Neogene sedimentary
depocentre (Fig. 1). An integrated age model is available based on a multidisciplinary
geochronological analysis of several boreholes within the SNS (Kuhlmann et al., 2006a,b)
and dinocyst biostratigraphy. The magnetostratigraphy, core correlation and age-diagnostic
dinocyst events used for this age-model are summarized in Fig. 2 and Table S1. The
recovered material mainly consists of fine-grained, soft sediments (clayey to very fine sandy),
sampled from cuttings, undisturbed sidewall cores and core sections (Fig. 2). Geochemical
analyses were limited to the (sidewall) core intervals, while the cuttings were to increase
resolution of the palynological samples, and are based on larger rock chips that have been
cleaned before treatment. Clear cyclic variations in the gamma ray signal and associated
seismic reflectors across the interval can be correlated across the entire basin (Kuhlman et al.,
2006a; Kuhlmann and Wong, 2008; Thöle et al. 2014). Samples from the two boreholes were
spliced based on the gamma-ray logs (Figs. 2, S2) and biostratigraphic events to generate a
composite record. The age model is mainly based on continuous paleomagnetic logging
supported by discrete sample measurements and high-resolution biostratigraphy. There is
evidence of small hiatuses above (~2.1 Ma) and significant hiatuses below the selected
interval (within the early Pliocene and Miocene, particularly the Mid-Miocene
Unconformity), which is why we excluded these intervals in this study. The position of the
Gauss-Matuyama transition at the base of log unit 6 correlates to the base of MIS 103, the
identification of the X-event, at the top of log unit 9, correlates to MIS 96, and the Olduvai
magnetochron is present within log units 16-18 (Kuhlmann et al., 2006a,b). These ages are
supported by dinocyst and several other bioevents (Table S1, updated from Kuhlmann et al.,
2006a,b). Consistent with the position of the X-event, the depositional model by Kuhlmann
and Wong (2008) relates the relatively coarse-grained, low gamma ray intervals to
interglacials characterized by high run off. A recent independent study on high-resolution
stable isotope analyses of benthic foraminifera from an onshore section in the same basin
confirmed this phase relation (Noorbergen et al., 2015). Around glacial terminations, when
sea level was lower but the basin remained fully marine, massive amounts of very fine-
grained clayey to fine silty material were deposited in the basin, the waste-products of intense
glacial erosion. During interglacials with high sea level more mixed, coarser-grained
sediments characterize the deposits, also reflecting a dramatically changed hinterland,
retreated glaciers, and possibly (stronger) bottom currents (Kuhlmann and Wong, 2008).
Based on this phase relation, detailed magneto- and biostratigraphy, grain size measurements,
and previous low resolution relative SST indices (Kuhlmann et al., 2004; Kuhlmann et al.,
2006a,b), the finer grained units are consistently correlated to MIS 102 – 92. Based on this
correlation of the GR inflection points to the corresponding LR04 MIS transitions, the
sequence is here transferred to an age scale through interpolation with a smoothing spline
function (Fig. S3).

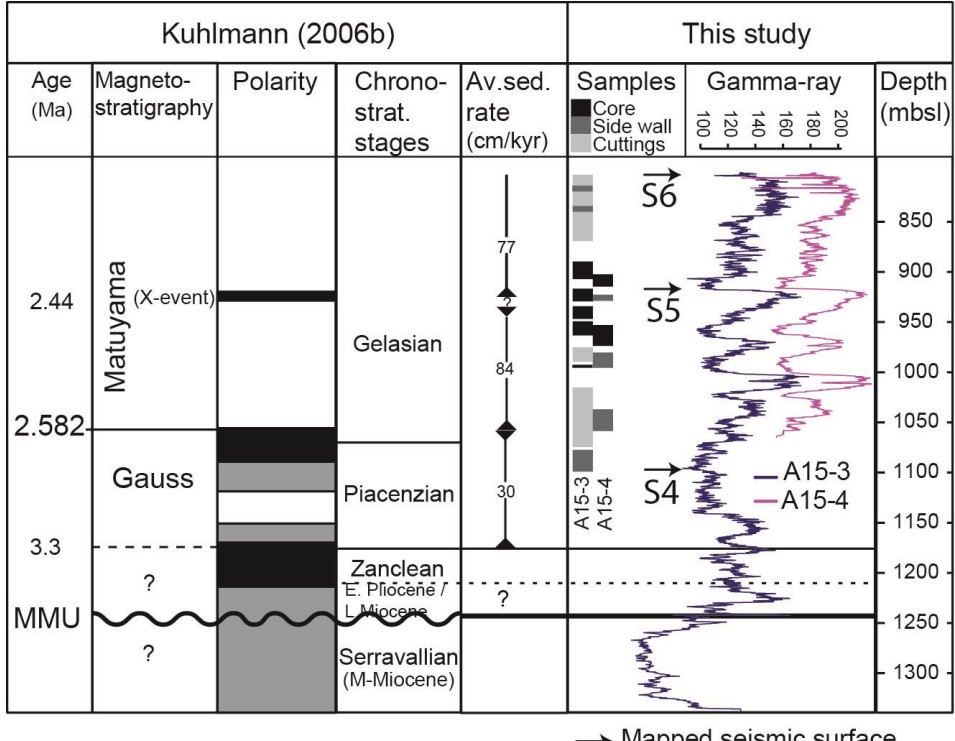


*Figure 2: Chronology and mean sedimentation rates as derived from biostratigraphy and paleomagnetic data (Kuhlmann et al., 2006a,b) in combination with the gamma-ray log of A15-3 and A15-4 used in this study on a common depth scale. The position of various sample types and the mapped seismic horizons S4-6 (Fig. S1) are indicated. Material for the sidewall cores is limited, and used only for palynology and organic geochemistry. Bioevents based on Kuhlmann et al. (2006a,b) are listed in Table S1.*

199

The regional structure and development of the delta front across the Plio-Pleistocene transition interval is very well constrained by a high-resolution regional geological model that represents the anatomy of the Eridanos (pro-) delta (Kuhlmann and Wong, 2008; Ten Veen et al., 2013). A total of 25 seismic horizons in the Plio-Pleistocene transition interval were mapped using series of publically available 2D and 3D seismic surveys across the northern part of the Dutch offshore sector. For all these surfaces the distribution of delta elements such as of topset-, foreset- and toeset-to-prodelta has been determined, resulting in zonal maps

(250 m grid size) that represent the present day geometry of the surfaces. The
paleoenvironmental reconstructions are compared to these maps to constrain the regional
setting and aid the interpretations.

**4 Paleoenvironmental proxies and methods**
*4.1 Benthic oxygen and carbon isotopes ($\delta^{18}O_b$ and $\delta^{13}C_b$)*
Oxygen and carbon isotopes were measured on tests of *Cassidulina teretis*, a cold water
species of endobenthic foraminifera that is generally abundant in the samples and common in
fine-grained sediment and relatively low salinities (Mackensen and Hald, 1988; Rosoff and
Corliss, 1992). Because of their endobenthic habitat, they record isotope compositions of pore
waters, which leads to somewhat reduced ($\delta^{13}C_b$) values compared to the overlying bottom
waters. Since the amount of material from the sidewall cores is limited, the isotope data is
only produced for the cored intervals with the principal aim to confirm the phase relationship
described by Kuhlmann and Wong (2008) between facies and climate. Preservation was based
on a visual inspection and assignment of a relative preservation scale of 1-5, after which the
poorest 2 classes were discarded because primary calcite was nearly absent. The best
preserved specimens (cat. 1) had shiny tests (original wall calcite) and showed no signs of
overgrowth. Category 2 specimens showed signs of overgrowth but were not recrystallized
and cat. 3 specimens were dull and overgrown by a thin layer of secondary calcite. Between
~20 and 50 μg of specimens per sample was weighed after which the isotopes of the
carbonate were measured using a Kiel III device coupled to a 253 ThermoFinnigan MAT
instrument. Isotope measurements were normalized to an external standard 'NBS-19' ($\delta^{18}O =$
-2.20‰, $\delta^{13}C = 1.95$‰)..

*4.2 Palynological proxies*
In modern oceans, dinoflagellates are an important component of the (phyto-)plankton. About
15-20% of the marine dinoflagellates form an organic walled cyst (dinocyst) during the life
cycle that can be preserved in sediments (Head, 1996). Dinocyst distribution in marine
surface sediments has shown to reflect changes in the sea surface water properties, mostly
responding to temperature (e.g., Rochon et al., 1999; Zonneveld et al., 2013). Down-core
changes in dinocyst assemblages are widely used in reconstructing past environmental
changes in the Quaternary (e.g., de Vernal et al., 2009), but also in the Neogene and
Paleogene (e.g., Versteegh and Zonneveld, 1994; Head et al., 2004; Pross and Brinkhuis,
2005; Sluijs et al., 2005; Schreck et al., 2013; De Schepper et al., 2011; 2013; Hennissen et
al., 2017).

Here we use the preference of certain taxa to cold-temperate to arctic surface waters to derive
sea surface temperature (SST) trends. The cumulative percentage of the dinocysts *Filisphaera*
*microornata, Filisphaera filifera, Filisphaera sp., Habibacysta tectata and B. tepikiense* on
the total dinocysts represents our cold surface water indicator (Versteegh and Zonneveld,
1994; Donders et al., 2009; De Schepper et al., 2011). Interestingly, *Bitectatodinium*
*tepikiense*, the only extant dinocyst among our cold-water species, has been recorded from the
mixing zone of polar front oceanic waters with cold brackish meltwaters from glacier ice
(e.g., Bakken and Dale, 1986) and at the transition between the subpolar and temperate zones
(Dale, 1996). The combined abundance of *Lingulodinium machaerophorum,*
*Tuberculodinium vancampoae*, *Polysphaeridium zoharyi* and *Operculodinium israelianum* is
used here to indicate, coastal waters, although they generally also relate to warmer conditions.
In particular, high percentages of *L. machaerophorum* are typically recorded in eutrophic
coastal areas where reduced salinity and (seasonal) stratification due to runoff occur (Dale,
1996; Sangiorgi and Donders, 2004; Zonneveld et al., 2009). At present, *T. vancampoae*, *P.*
*zoharyi* and *O. israelianum* are also found in lagoonal euryhaline environments (Zonneveld et
al., 2013), and hence could be used to indicate a more proximal condition relative to *L.*
*machaerophorum* (Pross and Brinkhuis, 2005).

At present, Protoperidinioid (P) cysts are mostly formed by heterotrophic dinoflagellates and
the percentage of P-cysts may be used as indicator of high eukaryotic productivity (cf.
Reichart and Brinkhuis, 2003; Sangiorgi and Donders, 2004; Sluijs et al., 2005). Here we use
the percentage of P-cysts (*Brigantedinium* spp., *Lejeunecysta* spp., *Trinovantedinium*
*glorianum*, *Selenopemphix* spp., *Islandinium* spp., *Barssidinium graminosum*, and *B. wrennii*)
to indicate eukaryotic productivity.

Terrestrial palynomorphs (sporomorphs) reflect variations in the vegetation on the
surrounding land masses and provide information on climate variables such as continental
temperatures and precipitation (e.g. Heusser and Shackleton, 1979; Donders et al., 2009;
Kotthoff et al., 2014). A ratio of terrestrial to marine palynomorphs (T/M ratio) is widely used
as a relative measure of distance to the coast and thereby reflects sea-level variations and
depth trends in the basin (e.g. McCarthy and Mudie, 1998; Donders et al., 2009; Quaijtaal et
al., 2014; Kotthoff et al., 2014). Morphological characteristics of late Neogene pollen types
can, in most cases, be related to extant genera and families (Donders et al., 2009; Larsson et
al., 2011; Kotthoff et al., 2014). In A15-3/4, the relatively long distance between the land and
the site of deposition means that the pollen assemblage is not only a reflection of vegetation
cover and climate, but includes information on the mode of transport. Assemblages with a
relatively high number of taxa, including insect pollinated forms, are indicative of substantial
pollen input through water transport (Whitehead, 1983), whereas wind-transported pollen
typically show a low-diversity. Sediments of a location proximal to a river delta likely receive
a majority of pollen that is water-transported, while distal locations are dominated by wind-
transported pollen and particularly bisaccate taxa (Hooghiemstra, 1988; Mudie and McCarthy,
1994). To exclude these effects, the percentage of arboreal pollen (AP), representing relative
terrestrial temperatures, was calculated excluding bisaccate forms. The non-arboreal pollen
(NAP; mainly Poaceae and also *Artemisia*, Chenopodiaceae and Asteraceae) consist only of
non-aquatic herbs. High AP percentages indicate warm, moist conditions, whereas open
vegetation (NAP and Ericaceae) is indicative for cooler, drier conditions consistent with a
glacial climate (Faegri et al, 1989).

*4.3 Palynological processing*
The samples were processed using standard palynological procedures (e.g., Faegri et al.,
1989) involving HCl (30%) and cold HF (40%) digestion of carbonates and silicates.
Residues were sieved with 15 μm mesh and treated by heavy liquid separation (ZnCl, specific
gravity 2.1 g/cm$^3$). The slides were counted for dinocysts (with a minimum of 100 cysts) and
pollen (with a preferable minimum of 200 grains). The dinocyst taxonomy follows Williams
et al. (2017). Resulting counts were expressed as percent abundance of the respective
terrestrial or marine groups of palynomorphs.

*4.4 Organic geochemical proxies*

301       We applied three measures for the relative marine versus terrestrial hydrocarbon

sources. The Carbon Preference Index (CPI), based on C$_{25}$-C$_{34}$ *n*-alkanes, originally devised to
infer thermal maturity (Bray and Evans, 1961), has high values for predominantly terrestrial
plant sources (Eglinton and Hamilton, 1967; Rieley et al., 1991). Values closer to one indicate
greater input from marine microorganisms and/or recycled organic matter (e.g., Kennicutt et
al., 1987). Furthermore, peat mosses like *Sphagnum* are characterized by a dominance of the
shorter $C_{23}$ and $C_{25}$ n-alkanes (e.g. Baas et al., 2000; Vonk and Gustafsson, 2009), whereas
longer chain n-alkanes ($C_{27}$-$C_{33}$) are synthesized by higher plants (e.g., Pancost et al., 2002;
Nichols et al., 2006) . Here we express the abundance of *Sphagnum* relative to higher plants
as the proportion of $C_{23}$ and $C_{25}$ relative to the $C_{27}$-$C_{33}$ odd-carbon-numbered n-alkanes.
Finally, the input of soil organic matter into the marine environment was estimated using the
relative abundance of branched glycerol dialkyl glycerol tetraethers (brGDGTs), produced by
bacteria that are abundant in soils, versus that of the marine Thaumarchaeota-derived
isoprenoid GDGT crenarchaeol (Sinninghe Damsté et al., 2002), which is quantified in the
Branched and Isoprenoid Tetraether (BIT) index (Hopmans et al., 2004). The distribution of
brGDGTs in soils is temperature dependent (Weijers et al., 2007; Peterse et al., 2012). Annual
mean air temperatures (MAT) were reconstructed based on down-core distributional changes
of brGDGT and a global soil calibration that uses both the 5- and 6-methyl isomers of the
brGDGTs ($MAT_{mr}$; De Jonge et al., 2014a). Cyclisation of Branched Tetraethers (CBT)
ratios, was shown earlier to correlate with the ambient MAT and soil pH (Weijers et al., 2007;
Peterse et al., 2012). The much improved CBT' ratio (De Jonge et al., 2014a), which includes
the pH dependent 6-methyl brGDGTs, is used here to reconstruct soil pH. The Total Organic
Carbon (TOC) and total nitrogen measurements are used to determine the atomic C/N ratio
that in coastal marine sediments can indicate the dominant source of organic matter, with
marine C/N values at ~10 and terrestrial between 15 and 30 (Hedges et al., 1997).

*4.5 Organic geochemical processing*

328        Organic geochemical analyses were limited to the core and sidewall core samples. For

TOC determination ~ 0.3 g of freeze dried and powdered sediment was weighed, and treated
with 7.5 ml 1 M HCL to remove carbonates, followed by 4 h shaking, centrifugation and
decanting. This procedure was repeated with 12 h shaking. Residues were washed twice with
demineralised water dried at 40-50°C for 96 h after which weight loss was determined. ~15 to
20 mg ground sample was measured in a Fisons NA1500 NCS elemental analyzer with a
normal Dumas combustion setup. Results were normalized to three external standards (BCR,
atropine and acetanilide) analyzed before and after the series, and after each ten
measurements. % TOC was determined by %C x decalcified weight/original weight.

For biomarker extraction ca. 10 g of sediment was freeze dried and mechanically powdered.
The sediments were extracted with a Dichloromethane (DCM):Methanol (MeOH) solvent
mixture (9:1, v/v, 3 times for 5 min each) using an Accelerated Solvent Extractor (ASE,
Dionex 200) at 100°C and ca. 1000 psi. The resulting Total Lipid Extract (TLE) was
evaporated to near dryness using a rotary evaporator under near vacuum. The TLE then was
transferred to a 4 ml vial and dried under a continuous $N_2$ flow. A 50% split of the TLE was
archived. For the working other half elemental sulfur was removed by adding activated (in
2M HCl) copper turnings to the TLE in DCM and stirring overnight. The TLE was
subsequently filtered over $Na_2SO_4$ to remove the CuS, after which 500 ng of a $C_{46}$ GDGT
internal standard was added (Huguet et al., 2006). The resulting TLE was separated over a
small column (Pasteur pipette) packed with activated $Al_2O_3$ (2 h at 150°C). The TLE was
separated into an apolar, a ketone and a polar fraction by eluting with n-hexane : DCM 9:1
(v/v), n-hexane : DCM 1:1 (v/v) and DCM : MeOH 1:1 (v/v) solvent mixtures, respectively.
The apolar fraction was analyzed by gas chromatography (GC) coupled to a flame ionization
detector (FID) and gas chromatography/mass spectroscopy (GC/MS) for quantification and
identification of specific biomarkers, respectively. For GC, samples were dissolved in 55 μl
hexane and analyzed using a Hewlett Packard G1513A autosampler interfaced to a Hewlett
Packard 6890 series Gas Chromatography system equipped with flame ionization detection,
using a CP-Sil-5 fused silica capillary column (25 m x 0.32 mm, film thickness 0.12 μm),
with a 0.53 mm pre-column. Temperature program: 70°C to 130°C (0 min) at 20°C/min, then
to 320°C at 4°C/min (hold time 20 mins). The injection volume of the samples was 1 µl.
Analyses of the apolar fractions were performed on a ThermoFinnigan Trace GC ultra,
interfaced to a ThermoFinnigan Trace DSQ MS using the same temperature program, column
and injection volume as for GC analysis. Alkane ratios are calculated using peak surface areas
of the respective alkanes from the GC/FID chromatograms.

Prior to analyses, the polar fractions, containing the GDGTs, were dissolved in *n*-hexane :
propanol (99:1, v/v) and filtered over a 0.45 µm mesh PTFE filter (ø 4mm). Subsequently,
analyses of the GDGTs was performed  using ultra high performance liquid chromatography-
mass spectrometry (UHPLC-MS) on an Agilent 1290 infinity series instrument coupled to a
6130 quadrupole MSD with settings as described in Hopmans et al. (2016). In short,
separation of GDGTs was performed on two silica Waters Acquity UHPLC HEB Hilic
(1.7µm, 2.1mm x 150mm) columns, preceded by a guard column of the same material.
GDGTs were eluted isocratically using 82% A and 18% B for 25 mins, and then with a linear
gradient to 70% A and 30% B for 25 mins, where A is *n*-hexane, and B = *n*-
hexane:isopropanol. The flow rate was constant at 0.2 ml/min. The [M+H]$^+$ ions of the
GDGTs were detected in selected ion monitoring mode, and quantified relative to the peak
area of the $C_{46}$ GDGT internal standard.

**5 Results**
*5.1 Stable isotope data*
The glacial-interglacial range in *Cassidulina teretis* $\delta^{18}O$ ($\delta^{18}O_b$) is ~1‰ between MIS 98 and
97, and ~1.3‰ between MIS 95 and 94, but with considerably more variation in especially
MIS 95 (Fig. 3). The $\delta^{13}C_b$ data co-vary consistently with $\delta^{18}O_b$ and have a glacial-interglacial
range of ~1.1‰, besides one strongly depleted value in MIS 94 (-3.5‰). The MIS 95 $\delta^{13}C_b$
values are less variable than the $\delta^{18}O_b$, pointing to an externally forced signal in the latter. The
$\delta^{18}O_b$ confirms the relation between glacial stages and fine grained sediment as proposed by
Kuhlman et al. (2006a,b). Although the data are somewhat scattered, the A15-3/4 phase
relation to the sediment facies is in agreement with the high-resolution stable isotope benthic
foraminifera record of the onshore Noordwijk borehole (Noorbergen et al., 2015). The glacial
to interglacial ranges are very similar in magnitude with those reported by Sosdian and
Rosenthal (2009) for the North Atlantic, but on average lighter by ~0.5‰ ($\delta^{18}O_b$) and ~1.8‰
($\delta^{13}C_b$).

*5.2 Palynology*
Palynomorphs, including dinocysts, freshwater palynomorphs and pollen, are abundant,
diverse, and well-preserved in these sediments. Striking is the dominance by conifer pollen.
Angiosperm (tree) pollen are present and diverse, but low in abundance relative to conifers.
During interglacials (MIS 103, 99, 97, 95, and 93) the pollen record generally shows
increased and more diverse tree pollen (particularly *Picea* and *Tsuga*), and warm temperate
*Osmunda* spores, whereas during glacials (MIS 102, (100), 98, 96, and 94) herb and heath
pollen indicative of open landscapes are dominant (Fig S2). The % arboreal pollen (AP; excl.
bisaccate pollen) summarizes these changes, showing maximum values of >40% restricted to
just a part of the coarser grained interglacial intervals (Fig. 3). The percentage record of cold
water dinocysts is quite scattered in some intervals but indicates generally colder conditions
within glacial stages, and minima during %AP maxima (Fig. 3). After peak cold conditions
and a TOC maximum (see below), but still well within the glacials, the % Protoperidinoid
consistently increases. Some intervals (e.g., top of MIS 94) are marked by influxes of
freshwater algae (*Pediastrum* and *Botryococcus)*, indicating a strong riverine input, these data
however do not indicate a clear trend. This robust in-phase pattern of glacial-interglacial
variations is also reflected by high T/M ratios during glacials, indicating coastal proximity,
and low T/M during (final phases of) interglacials. The Glacial-Interglacial (G-IG) variability
in the T/M ratio is superimposed on a long-term increase. The coastal (warm-tolerant)
dinocyst maxima are confined to the interglacial intervals and their abundance increases
throughout the record. Successive increases of coastal inner neritic *Lingulodinium*
*machaerophorum*, followed by increases in coastal lagoonal species in the youngest part,
mirror the shoaling trend in the T/M ratio, which in time correspond with the gradual
progradation of the Eridanos delta front (Fig. S1).

*5.3 Organic geochemical proxies*
The lowest TOC contents are reached in the clay intervals, and typically range between 0.5%
in glacials and 1% in interglacials (Fig. 3). Nitrogen concentrations are relatively stable
resulting in C/N ratios primarily determined by organic carbon content, ranging between ~8-9
(glacials) and ~14 and 17 (interglacials). The Carbon Preference Index (CPI) is generally
high, reflecting a continuous input of immature terrestrial organic matter. Minimum CPI
values of ~2.8 - 2.9 are reached at the transitions from the coarser sediments to the clay
intervals after which they increase to maxima of 4.5 - 5.0 in the late interglacials. The $n$-C$_{23+25}$
*Sphagnum* biomarker correlates consistently with the T/M ratio, %AP, and cold water
dinocysts (Fig. 3), while the variation in the CPI index is partially out of phase; it is more
gradual and lags the % TOC and other signals. Generally lower Branched and Isoprenoid
Tetraether (BIT) index values during interglacials (Fig. 3) indicate more marine conditions,
i.e. larger distance to the coast and relatively reduced terrestrial input from the Eridanos
catchment (cf. Sinninghe Damsté, 2016). As both brGDGT input (run off, soil exposure and
erosion) and sea level (distance to the coast) vary across G-IG timescales, for example during
deglaciation and subsequent reactivation of fluvial transport (Bogaart and van Balen, 2000),
the variability of the BIT index is somewhat different compared to the T/M palynomorph ratio
(Fig. 3), but is generally in phase with gradual transitions along G-IG cycles. The $MAT_{mr}$-
based temperature reconstructions vary between 5 and 17$^{\circ}$C, reaching maximum values in
MIS 97. However, in the MIS 99/98 and MIS 96/95 transitions the $MAT_{mr}$ shows variability
opposite to the identified G-IG cycles and the signal contains much high-order variability.
Low values during interglacials generally coincide with low CBT'-reconstructed soil pH of
<6.0 (Fig. 3).

*Figure 3: spliced record of A15-3 and A15-4 showing the principal geochemical and*
*palynological indices. Shaded blue intervals represent the identified glacial MIS delimited by*
*the gamma-ray transitions following Kuhlmann et al. (2006a,b). Data density is dependent on*
*type of sample as indicated in Fig. 1. Age scale is based on correlation and LOESS*
*interpolation of the identified MIS transitions to the LR04 benthic stack (Lisiecki and Raymo,*
*2005) as shown in Fig S3. Data is available in Tables S2 and 3.*

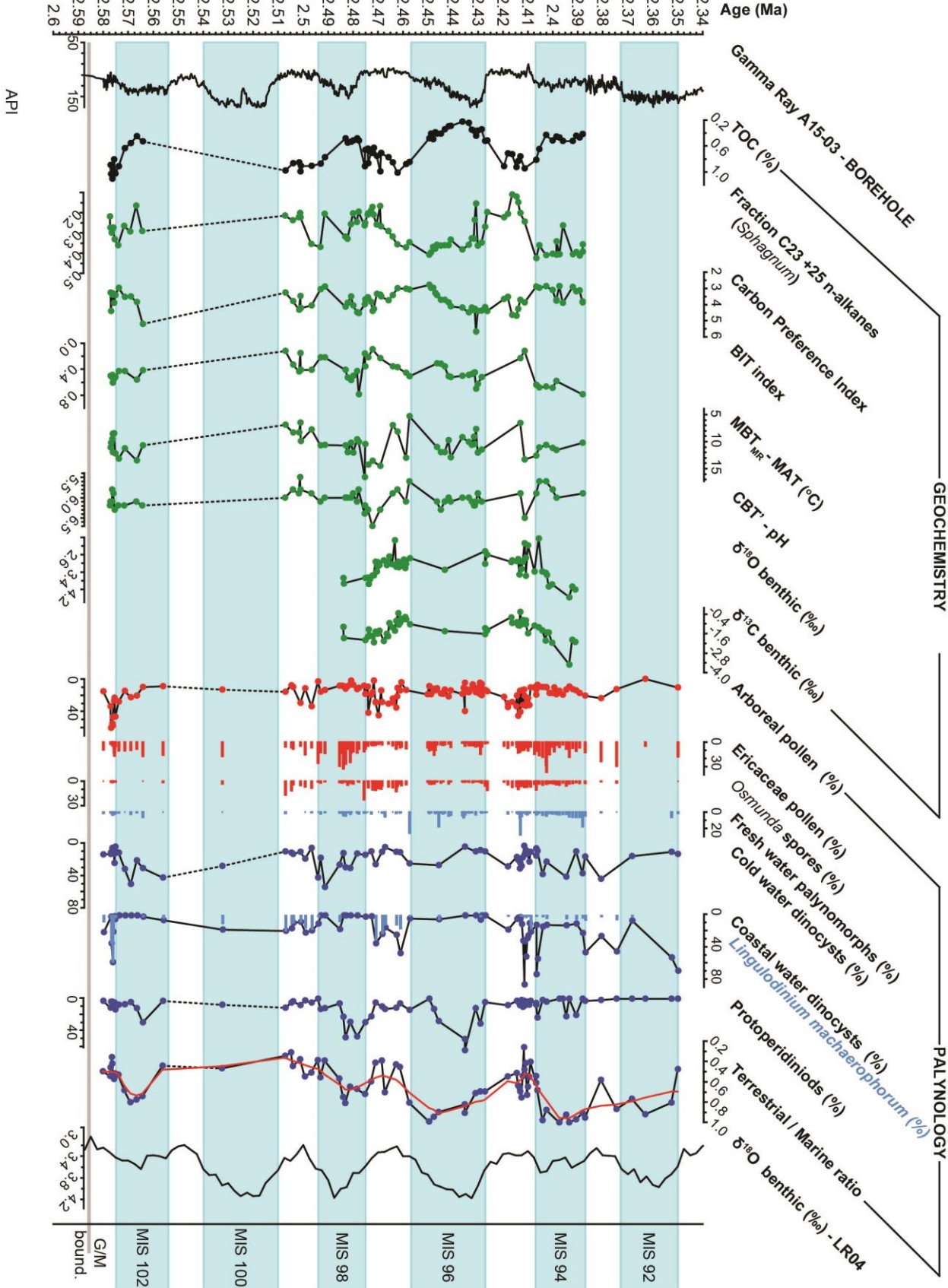

**6 Discussion**

*6.1 Paleoenvironmental setting and climate signals*

The source area study by Kuhlmann et al. (2004) indicated the Eridanos paleoriver as the principal source of the terrestrial deposits. The detailed seismic interpretations indeed show the advancing Eridanos delta front from the east toward the sites (Fig. S1). This trend is captured by the long-term increases in the T/M ratio and the proportion of coastal dinocysts (Fig. 3). In- or exclusion of bisaccate pollen in the T/M index (Fig. S5), the component most sensitive to differential transport processes, indicates no direct influence of differential transport on the T/M ratio. During MIS 103, 99, 97, 95, and 93 the AP% increases indicate generally warmer and more humid conditions than during MIS 102, 98, 96, and 94 (Fig. 3). The cold-water temperature signal based on dinocysts is more variable than the terrestrial cooling signals from the AP%. Pollen assemblages represent mean standing vegetation in the catchment, and also depend on dominant circulation patterns and short-term climate variations (Donders et al., 2009). Due to exclusion of bisaccate pollen, the %AP is generally low but eliminates any climate signal bias due to the direct effect of sea level changes (Donders et al., 2009; Kotthoff et al., 2014). In the record there are small but significant time lags between proxies, which have important implications for explaining the forcing of G-IG cycles. In the best constrained MIS transition (98 to 97), the G-IG transition is seen first in decreases of the cold water dinocysts and $n$-$C_{23+25}$ n-alkanes predominantly derived from *Sphagnum*. Subsequently the BIT decreases, and $MAT_{mr}$ and the %AP increase, and finally the $\delta^{18}O_b$ and T/M ratio decrease with a lag of a few thousand years (Fig. 3). Changes in the CPI record are more gradual, but generally in line with T/M. The AP% and T/M proxies have the most extensive record and detailed analysis of several glacial-interglacial transitions shows that the declines in AP% consistently lead the T/M increases by 3-8 kyr based on the present age model (Fig. S2). The T/M ratio variability corresponds well to the LR04 benthic stack (Fig.

3), which is primarily an obliquity signal. Within the constraints of the sample availability,
our record captures the approximate symmetry between glaciation and deglaciation typical of
the Early Pleistocene (Lisiecki and Raymo, 2005).

The high variability and strongly depleted values in $\delta^{18}O_b$ during MIS 95 occur during peak
coastal dinocyst abundances, suggesting high run off during maximum warming phases.
During cold water dinocysts maxima, the high abundance of Protoperidinioids indicates high
nutrient input, and productive spring/summer blooms, which point to strong seasonal
temperature variations. This productivity signal markedly weakens in MIS 94 and 92 and the
gradual T/M increase is consistent with the basin infill and gradually approaching shelf-edge
delta (Fig. S1). As Protoperidinioid minima generally occur during TOC maxima there is no
indication for a preservation overprint since selective degradation typically lowers relative
abundances of these P-cysts (Gray et al., 2017). Combined, the high TOC and CPI values,
coastal and stratified water conditions, and intervals of depleted $\delta^{18}O_b$ document increased
Eridanos run-off during interglacials. These suggest a primarily terrestrial organic matter
source that, based on mineral provenance studies (Kuhlmann et al., 2004) and high conifer
pollen abundance documented here, likely originated from the Fennoscandian Shield. The
fine-grained material during cold phases is probably transported by meltwater during summer
from local glaciers that developed since the late Pliocene at the surrounding Scandinavian
mainland (Mangerud et al., 1996; Kuhlmann et al., 2004).

*6.2 Temperature reconstruction and brGDGT input*
Whereas the BIT index reflects the G-IG cycles consistently, the $MAT_{mr}$ record, which is
based on GDGTs, has a variable phase relation with the G-IG cycles and high variability. The

use of $MAT_{mr}$ in coastal marine sediments is based on the assumption that river-deposited

brGDGTs reflect an integrated signal of the catchment area. As the Eridanos system is

reactivated following glacials, glacial soils containing brGDGT are likely eroded causing a

mixed signal of glacial and interglacial material. The lowest $MAT_{mr}$ and highest variability is

indeed observed during periods of deposition of sediments with a higher TOC content and

minima of CBT'-derived pH below 6 (Fig. 3), consistent with increased erosion of acidic

glacial (peat) soil. Additional analysis of the apolar fractions in part of the samples reveals

during these periods a relatively high abundance of the $C_{31}$ 17α, 21β-homohopanes, which in

immature soils indicates a significant input of acidic peat (Pancost et al., 2002). This suggests

that the variability in the $MAT_{mr}$ record is not fully reliable due to (variable) erosion of glacial

soils or peats. Alternatively, the terrestrial brGDGT signal may be altered by a contribution of

brGDGTs produced in the marine realm. BrGDGTs were initially believed to be solely

produced in soils, but emerging evidence suggests that brGDGTs are also produced in the

river itself (e.g., Zell et al., 2013; De Jonge et al., 2014b) and in the coastal marine sediments

(e.g., Peterse et al., 2009; Sinninghe Damsté, 2016 Based on the modern system, the degree of

cyclisation of tetramethylated brGDGTs ($\#rings_{tetra}$) has been proposed to identify a possible

in situ overprint (Sinninghe Damsté, 2016). The $\#rings_{tetra}$ in this sediment core is <0.37,

which is well below the suggested threshold of 0.7, and thus suggests that the brGDGTs are

primarily soil-derived. However, a ternary diagram of the brGDGT distribution show some

offset to the global soil calibration that decreases with increasing BIT values (Fig. S6),

pointing to some influence of in-situ GDGT production when terrestrial input is relatively

low. Finally, selective preservation in the catchment and during fluvial transport may have

affected the brGDGT signal, although experimental evidence on fluvial transport processes

indicates that these do not significantly affect initial soil-brGDGT compositions (Peterse et

al., 2015).

525

*6.3 Implications for the intensification of Northern Hemisphere glaciations*

The classic Milankovitch model predicts that global ice volume is forced by high northern

summer insolation (e.g. Hays et al., 1976). Raymo et al. (2006) suggested an opposite

response of ice sheets on both hemispheres due to precession forcing, cancelling out the

signal and amplifying obliquity in the early Pleistocene. That hypothesis predicts that regional

climate records on both hemispheres should contain a precession component that is not visible

in the sea level and deep sea $\delta^{18}O_b$ record, and is supported by evidence from Laurentide Ice

Sheet melt and iceberg-rafted debris of the East Antarctic ice sheet (Patterson et al., 2014;

Shakun et al., 2016). Alternatively, a dominantly obliquity forced G-IC cycle is supported by

a significant temperature component in the temperature deep sea $\delta^{18}O_b$ record (Sosdian and

Rosenthal, 2009) and dominant 41-kyr variability in North American biomarker dust fluxes.

Our results show that the regional NH climate on both land and sea surface vary on the same

timescale as the local relative sea level which, with the best possible age information so far

(Fig. S3), mirrors the global LR04 $\delta^{18}O_b$ record. The temperature changes lead the local sea

level by 3-8 kyr, which is consistent with a NH obliquity forcing scenario as cooling would

precede ice buildup and sea level change. Contrary to the model proposed by Raymo et al.

(2006), this suggests that the NH obliquity forcing is the primary driver for the glacial-

interglacial in the early Pleistocene, although we cannot exclude precession forcing as a

contributing factor.. Various studies indicate the importance of gradual $CO_2$ decline in the

intensification of NHG (Kürschner et al., 1996; Seki et al., 2010; Bartoli et al., 2011)

combined with the threshold effects of ice albedo (Lawrence et al., 2010; Etourneau et al.,

2010) and land cover changes (Koenig et al., 2011). Simulations of four coupled 3-D ice

models indicate that Antarctic ice volume increases respond primarily to sea-level lowering,

while Eurasian and North American ice sheet growth is initiated by temperature decrease (de

Boer et al., 2012). The latter dominate the eustatic sea-level variations during glacials. Our
observations agree with the modelled temperature sensitivity of NH ice sheet growth. The
dominant obliquity signal further suggests a seasonal aspect of the climate forcing. The
combination of high summer productivity, based on increased Protoperidiniod dinocysts, and
increased proportions of cold dinocysts during the glacials in the SNS record indicate a strong
seasonal cycle. This confirms similar results from the North Atlantic (Hennissen et al., 2015)
and is consistent with an obliquity-driven glacial-interglacial signal in a mid-latitudinal
setting, likely promoting meridional humidity transport and ice buildup.

The southward migration of Arctic surface water masses indicated by increases in  cold water
dinocysts (Fig. 3) is furthermore relevant for understanding the relation between the Atlantic
meridional overturning circulation (AMOC) intensity and ice sheet growth (e.g. Bartoli et al.,
2005; Naafs et al, 2010). Mid-Pliocene increased heat transport and subsequent decrease
during NHG due to AMOC intensity changes has been invoked from many proxy records but
is difficult to sustain in models (Zhang et al., 2013). Our results indicate that the NW
European early Pleistocene climate experienced significant cooling in all temperature-
sensitive proxies during sea-level lowstands, which is consistent with southward displacement
of the Arctic front and decreased AMOC (Naafs et al., 2010). The $MAT_{mr}$ indicates a 4-6 $^{o}$C
glacial-interglacial amplitude although the timing is offset relative to the other proxies. The
data-model mismatch in AMOC changes might be due to dynamic feedbacks in vegetation or
(sea-) ice (Koenig et al., 2011; de Boer et al., 2012) that are prescribed variables in the model
comparison by Zhang et al. (2013).

In addition, our SNS record provides a well-dated early Pleistocene Glacial-Interglacial
succession integrating marine and terrestrial signals improving on the classic terrestrial
Praetiglian stage. While conceptually valid, the earliest Pleistocene glacial stages defined in
the continental succession of the SE Netherlands (Van der Vlerk and Florschütz, 1953;
Zagwijn, 1960) and currently considered text book knowledge, are highly incomplete and
locally varied (Donders et al., 2007). This shallow marine SNS record provides a much more
suitable reflection of large-scale transitions and trends in NW Europe and merits further
development by complete recovery of the sequence in a scientific drilling project (Westerhoff
et al., 2016).

**7 Conclusions**
The independently dated late Pliocene-early Pleistocene sedimentary succession of the
southern North Sea Basin provides a record that straddles the intensification of Northern
Hemisphere Glaciation and the subsequent climate fluctuations in a shallow marine setting in
great detail. The intensification of the glaciation and the correlation to marine isotope stages
103 to 92, including the conspicuous first Pleistocene glacial stages 98, 96 and 94, is well
expressed in the marine and terrestrial palynomorph and organic biomarker records of the
southern North Sea. The independent relative sea- and land-based temperature records show
clearly coeval (at this resolution) expression of glacial-interglacial and sea-level cycles that
are well-correlated to the LR04 benthic stack. Critically, both the biomarker signals, %AP,
and cold water dinocyst variations show consistent in-phase variability on obliquity time
scales, leading sea-level changes by 3-8 kyr, which supports a dominantly direct NH
insolation control over early Pleistocene glaciations. Based on this integrated record, NH
obliquity forcing is the primary driver for the glacial-interglacial cycles in the early
Pleistocene. Furthermore, our findings support the hypothesis of temperature sensitivity of
NH ice sheet growth. The interglacials are characterized by (seasonally) stratified waters
and/or near-shore conditions as glacial-interglacial cycles became more expressive and the
Eridanos delta progressed into the region. The strong seasonality at mid-latitudes point to a
vigorous hydrological cycling that should be considered as a potential factor in ice sheet
formation in further investigations.

## 8 Author contributions

THD, HB and GK designed the research. NvH carried out the geochemical analyses under
supervision of JW, GJR, FP and JSSD. RV, DM and THD carried out the palynological
analyses and interpreted the data together with FS. LL and RPS provided stable isotope data
on benthic foraminifera. JtV provided seismic interpretations. THD integrated the data and
wrote the paper with contributions from all authors.

## 9 Acknowledgements

We are grateful the constructive comments of Stijn de Schepper and David Naafs and an
anonymous referee that helped to improve the manuscript. We gratefully acknowledge the
support in providing the offshore samples to this study and permission to publish by
Wintershall Noordzee B.V., and project support by partners Chevron Exploration and
Production Netherlands B.V., Total E&P Nederland B.V., Dana Petroleum Netherlands B.V.,
Oranje-Nassau Energie B.V., and Energie Beheer Nederland (EBN). Arnold van Dijk is
thanked for running C/N and stable isotope analyses, and Giovanni Dammers for processing
palynological samples. The work was partly supported by funding from the Netherlands Earth
System Science Center (NESSC) through a gravitation grant (NWO 024.002.001) from the
Dutch Ministry for Education, Culture and Science to JSSD, GJR, and LL.

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

Contributions Series 48, College Station, TX, 1097 pp. 97–108
Van der Vlerk, I.M, Florschütz, F. 1953. The palaeontological base of the subdivision of the
Pleistocene in the Netherlands. Verhandelingen Koninklijke Nederlandse Akademie van
Wetenschappen, Afdeling Natuurkunde, 1e Reeks XX(2): 1–58.
Versteegh, G.J.M., Zonneveld, K.A.F., 1994. Determination of (palaeo-)ecological
preferences of dinoflagellates by applying detrended and canonical correspondence analysis
to late Pliocene dinoflagellate cyst assemblages of the south Italian Singa section. Review of
Palaeobotany and Palynology 84: 181–199. https://doi.org/10.1016/0034-6667(94)90050-7.
Vonk, J.E., Gustafsson, Ö, 2009. Calibrating n-alkane *Sphagnum* proxies in sub-Arctic
Scandinavia. Organic Geochemistry 40: 1085-1090.
https://doi.org/10.1016/j.orggeochem.2009.07.002.
Zagwijn, W.H., 1960. Aspects of the Pliocene and early Pleistocene vegetation in The
Netherlands. Mededelingen van de Geologische Stichting, Serie C III-1–5, 1–78.
Zell, C., Kim, J.-H., Moreira-Turcq, P., Abril, G., Hopmans, E.C., Bonnet, M.-P., Sobrinho,
R. L., and Sinninghe Damsté, J.S., 2013. Disentangling the origins of branched tetraether
lipids and crenarchaeol in the lower Amazon River: implications for GDGT-based proxies,
Limnology and Oceanography. 58, 343–353. DOI: 10.4319/lo.2013.58.1.0343.
Zhang, Z.-S., Nisancioglu, K. H., Chandler, M. A., Haywood, A. M., Otto-Bliesner, B. L.,
Ramstein, G., Stepanek, C., Abe-Ouchi, A., Chan, W.-L., Bragg, F. J., Contoux, C., Dolan, A.
M., Hill, D. J., Jost, A., Kamae, Y., Lohmann, G., Lunt, D. J., Rosenbloom, N. A., Sohl, L. E.,
and Ueda, H., 2013. Mid-pliocene Atlantic Meridional Overturning Circulation not unlike
modern. Climate of the Past 9, 1495-1504.DOI :10.5194/cp-9-1495-2013
Ziegler, P.A., 1990. Geological Atlas of Western and Central Europe (2$^{nd}$ edition).Shell
Internationale  Petroleum  Maatschappij  B.V.;  Geological  Society Publishing House (Bath),
239 pp.
Zöllmer, V. and Irion, G., 1996. Tonminerale des Nordseeraumes ihr Verteilungsmuster in
kreidezeitlichen bis pleistozänen Sedimentabfolgen und in den Oberflächensedimenten der
heutigen Nordsee: Courier Forschungsinstitut Senckenberg, 190. Frankfurt am Mainz, 72 p.
Zonneveld, K.A.F., Marret, F., Versteegh, G.J.M., Bogus, K., Bonnet, S., Bouimetarhan, I.,
Crouch, E., de Vernal, A., Elshanawany, R., Edwards, L., Esper, O., Forke, S., Grøsfjeld, K.,
Henry, M., Holzwarth, U., Kielt, J.-F., Kim, S.-Y., Ladouceur, S., Ledu, D., Chen, L.,
Limoges, A., Londeix, L., Lu, S.-H., Mahmoud, M.S., Marino, G., Matsouka, K.,
Matthiessen, J., Mildenhal, D.C., Mudie, P., Neil, H.L., Pospelova, V., Qi, Y., Radi, T.,
Richerol, T., Rochon, A., Sangiorgi, F., Solignac, S., Turon, J.-L., Verleye, T., Wang, Y. &
Young, M., 2013. Atlas of modern dinoflagellate cyst distribution based on 2405 data points.
Review of Palaeobotany and Palynology 191: 1-197.
https://doi.org/10.1016/j.revpalbo.2012.08.003.