# Peer review of "dominant Northern Hemisphere forcing"

_Climate of the Past, 2017_

## Referee Comment (RC1) · S. De Schepper (Referee) · 26 Oct 2017

The authors set out to test the phase relation between forcing and climate response around the Plio-Pleistocene transition. They use a composite record from the North Sea Basin and present new palynological and geochemical records to document the marine and terrestrial climate evolution. The site in a shallow marine sea is ideally suited to couple the climate signal from both realms. The authors identify variations in their data between G and IG, and conclude that NH obliquity forcing is the main driver for G-IG cycles in the Early Pleistocene.

I have some concerns about the age model (pmag, bioevents), the identification of

leads/lags, use of cutting samples in paleoclimate studies and the environmental interpretation of the dinocysts.

Age model.

Discussing phase relations between land, ocean and ice sheets hinges crucially on the age model. While the presented work is underpinned by previously published papers and insights into depositional environment (papers by Kuhlmann and co-authors), aspects of the age model can be questioned.

The authors rely here on the G/M reversal and the X-event for constraining the age of their studied interval (L152–155). Kuhlman and Wong (2008) discuss in fact 4 different possible interpretations of the pmag. It seems very questionable to me that the very short-lived X-event (2.420–2.441 Ma, Cande and Kent, 1995) can be detected in the sedimentary record of a shallow sea by measuring the magnetic signal of discrete samples (Kuhlman and Wong, 2008). This event does not show up in u-channeled, high-resolution pmag records of the North Atlantic (e.g. Hoddell and Channell 2016; Channell et al 2016), neither has it been tied to the LR04 Marine Isotope Stratigraphy. The dinocyst bioevents generally point to the Plio-Pleistocene, but the events are not well-recognised (e.g. Barssidinium, M. choanophorum) or not calibrated (e.g. I. multiplexum) outside the North Sea Basin. This questions the age assigned to these events and thus also the age model. Using additional/different tiepoints that have been calibrated outside the North Sea Basin could provide more credibility to the age model used (see below). Based on these concerns about the age model, it remains uncertain 1) whether the cycles visible in the gamma-ray reflect the interval MIS102–96 and 2) whether these are truly, consecutive (i.e. no erosional events in between) G-IG cycles.

Leads/lags.

The leads/lags between climate proxies and sea level are not so clearly visible as the authors claim in the abstract and conclusions. The leads/lags are not clearly demonstrated on a figure, or more importantly using statistical techniques. In fact, they are

Interactive
comment

only mentioned on L495–496 in the discussion. Furthermore, in L491-493, the authors write that "...regional NH climate on land and sea surface varies in concert with local relative sea level...". That, to me, suggests there are no leads/lags.

Detailed comments on the bioevents.

Several of the events used in Kuhlmann et al. 2006 (Palaeo-3) are ecologically controlled events (e.g. acmes, peaks in O. israelianum) that have not been detected outside the North Sea Basin. Even the LOD of I. multiplexum has not been calibrated to an independent time scale. Including these points as tiepoints thus make the age model questionable (Fig. S3). Generally, the assemblage does point to the Late Pliocene / Early Pleistocene. For example, the increase in H. tectata is a well-established event in the eastern North Atlantic at the Plio-Pleistocene transition (∼2.6 Ma, De Schepper and Head 2009; Hennissen et al. 2015). But the choice of other bioevents used can be questioned: 1. LOD Barssidinium spp.: Barssidinium pliocenicum ranges up to 2.0 Ma in Iceland (Verhoeven et al. 2011) and 1.95 Ma in St. Erth Beds (Head 1993). In the southern North Sea (Belgium) it extends into the Merksem Sands (∼2.6 Ma), but Early Pleistocene deposits are not present there. 2. LOD M. choanophorum. The HO is strongly diachronous in the Atlantic region, making this a bad species to use for biostratigraphy. It has a HO in Norwegian Sea ∼3.3 Ma, De Schepper et al. 2017); HCO around 3.0 Ma, but frequent records up to 2.75 Ma and rare occurrences in the Pleistocene of DSDP Hole 610 (De Schepper & Head 2009); HO in southern North Sea ∼3.2–2.7 Ma (De Schepper et al. 2009; Louwye et al. 2004, Louwye and De Schepper. 2010). In fact, this species is still around in the modern Gulf of Mexico (Limoges et al. 2013). Suggestions to improve the age model include using bioevents listed in De Schepper and Head 2008 (Stratigraphy): I. lacrymosa has a well-established and relatively synchronous HO in the eastern North Atlantic and Mediterranean in or immediately prior to MIS G4–G6. O.? eirikianum has a HO in ∼2.6 Ma (MIS 104–103) in the eastern North Atlantic, while in the subtropical North Atlantic it extends to 2.34 Ma (∼MIS 94) (De Schepper et al. 2008).

Cuttings.

What is the effect of using cutting samples (caving, reworking) on your interpretations? This should be discussed in more detail. Cuttings can introduce caved material and together with reworked material is extremely difficult to extract meaningful paleoclimatic and biostratigraphic signals. A detailed account sample type (cutting, SWC, core) for each proxy would be useful to assess the effect of cuttings on the records. Figure 2 does show the use of different sample types, but the effect on the proxies is not discussed adequately. For example, was the FOD of I. multiplexum identified in a cutting sample? Could it be an artefact of the drilling procedure (i.e. younger material introduced into older material due to caving)?

Environmental signal from dinocysts.

The 4 species used to indicate a warm water signal are all coastal, shallow water species (L225–227). Their distribution in the shallow North Sea Basin could be strongly affected by SL fluctuations at the beginning of the Early Pliocene. Versteegh (1994) therefore does not include these taxa in a warm-cool index. Furthermore, L. machaerophorum is often used to indicate river input and sea level fluctuations (Holzwarth et al. 2010). How do you disentangle the effect of sea level and temperature for these 4 species, when their distribution could be affected by both? The T/M ratio is interpreted as a relative SL indicator. While this intuitively seems correct, I wonder if the relation is that simple? Terrestrial palynomorphs are affected by transport patterns (wind, position of rivers) and could thereby influence the sea level interpretation?

Minor comments.

L45 There is hardly proxy data to say something about MIS 100.

L47 Freshwater flux is not really supported by the fresh water algae.

L50-51 Confusing. Please rephrase.

L52 SST is not a good indicator of migration of a watermass. Microfossil assemblage

could help you with identifying such migration, but not SST alone.

L73 space missing before "and"

L105 rephrase "but which stratigraphic position"

L110 During the Neogene, there could have been a southerly connection between the North Sea and Atlantic (see reconstructions of e.g. Gibbard and Lewin 2003, 2016). It might be worth to use the more recent Gibbard and Lewin 2016 palaeogeographic reconstruction in stead of Zeigler 1990 (Figure 1).

L121 "different water types": only reference to Atlantic water?

L123 blue, not black?

L126–128 Please provide a timeframe. Did this occur in the entire Neogene? Pliocene/Pleistocene only?

L157–162 This model would get more credibility if this has also been demonstrated for late Pleistocene glacial/interglacial cycles. Would the SL drop of up to 60 m in these glacials (e.g. Miller et al. 2005; Bintanja et al. 2005) not provide a stronger control on the sedimentation (rather than hydrography)?

L165 How was the age model transferred to LR04 MIS? This is not clear. Fig. S3 shows an age model, not the link with the LR04 stratigraphy.

L174–175 Please check also De Schepper et al. 2017.

L177–L186 Does this paragraph belong in the age model section?

L190 C. teretis in italics.

L202 How was recrystallization and dissolution determined? SEM analyses would be necessary. See the need for SEM in e.g. Risebrobakken et al. 2016 (Paleoceanography).

L211 de Vernal (no capital D)

**CPD**
[Figure]

L270 delete ", dinocysts"

L273 Why were there only relative abundances calculated? Typically, concentrations (and accumulation rates) provide support for your relative abundance based interpretations.

L304 Delete "For TOC determination"

L359 Not convincing when looking at Fig. 3, because the data is mainly restricted to MIS 97 and 95/94. There are very few data points in MIS 96 and 98, and none in MIS 100 – the three intervals where gamma-ray values are highest. L363 vary, not very

L372 diverse

L377 Are herb and heath pollen dominant? Pinus remains the dominant species. Please make clear that you are discussing the pollen record, excluding pine pollen.

L383 Which fresh water algae did you find?

L401-402 What does the n-C23 Sphagnum biomarker indicate in terms of the climate system/environment?

L413 MIS 96/95 (space missing)

L420-421 It needs to be better documented how LR04 MIS transitions are recognized (see earlier).

L422 Tables S and 2?

Fig. 3 The Lingulodinium machaerophorum record should be presented separately – difficult to see now.

L428–... Chapter 6.1 is confusing and does not really deal with paleoenvironment. It is not clear which MIS is discussed, and the switching between proxies (e.g. L429–433) and time intervals (all glacial/interglacials, MIS 98/97, 94 and 92) makes this difficult to follow.

L432 depend (not depends)

L434 Effect of SL on pollen is addressed here, but the effect of SL on the dinocyst record is not discussed in the MS.

L485, L535 Onset/intensification have been used intermixed. I think it is commonly accepted that the onset of NHG occurs at 3.6 Ma (e.g. Mudelsee and Raymo 2005) and that for the period around the Plio-Pleistocene transition the term intensification should be used. L492, 496 Please make a reference back to the proxy that is used to infer local relative sea level. If you are referring to the T/M ratio as your sea level proxy, do take into account that T is affected by different transport mechanisms (L251-258).

L495–496 What is small – please specify? Please indicate which figure shows the small lead.

L510 Speculation.

L518 Severe cooling. Subjective comment, certainly if you know that L. machaerophorum does not occur in regions with summer SST below 15ËŽC. This species is present in all glacials.

Fig. S3 Please provide a list with the tiepoints (event, type, age, age scale used).

---

## Referee Comment (RC2) · D. Naafs (Referee) · 10 Nov 2017

In this manuscript Donders et al. provide new proxy records from an early Pleistocene (MIS 103-92) marine sediment core from the North Sea. They use an impressive array of inorganic and organic proxies (e.g. pollen, dinoflagellates, brGDGTs) to reconstruct changes in paleoenvironment (temperature and vegetation) in the hinterland and regional surface water conditions (sea surface temperature). Based on this data they argue that low land and sea temperatures characterized glacial stages, while higher temperatures and influx of riverine freshwater characterized interglacials. The main conclusion is that these climatic changes appears to be obliquity paced and not

precession-drives as has been suggested before (Raymo et al., 2006).

I am not an expert in pollen or dinoflagellates and will therefore focus my review on the organic geochemistry and overall conclusions.

The main focus of the manuscript is the "proposed disputed on the phase relation between forcing and climatic response in the early Pleistocene". I agree with the authors that such a dispute exists and answering it is of importance for our understanding of the climate system. However, the discussion on this specific topic in this manuscript is rather limited and is missing a discussion of crucial prior work on this topic. This is important as without such detailed discussion on the global implications of the findings, the paper provides not much more than a regional climatic story, making it of much less interest to the diverse readership of climate of the past. I think this work should be published in climate of the past, but I suggest that the authors provide a much more thorough discussion and include key references on this topic such as (Shakun et al., 2016), (Naafs et al., 2012), and (Patterson et al., 2014) in the revised manuscript.

In addition, I wonder whether the age model is robust enough. The low-resolution benthic d18O record of this site does not always look like the LR04 stack. In particular, MIS 94 is characterized by very negative d18O values in the core, while this is not obvious in the LR04 stack. I urge the authors to explore the uncertainty of the age model more. In terms of looking at forcing of climate during the early Pleistocene, it does not really matter what MIS we look at, but for comparison with other proxy records it does.

Minor comments in order of appearance: Line 51-55: this is a bit of a weird ending of the abstract, especially in the context of the main focus of the paper that is stated at the beginning of the abstract. The authors should end the abstract with a clear conclusion of what, according to their work, the phase relation is between forcing and climatic response.

Line 66: a full review paper on IRD in the North Atlantic during the Plio/Pleistocene is

given in (Naafs et al., 2013)

Line 73-78: somewhere make reference to mechanism proposed in (Haug et al., 2005)

Line 82-88; here other recent publications that refute or support Raymo's hypothesis (e.g., Naafs et al., 2012; Patterson et al., 2014; Shakun et al., 2016) should be introduced to provide a clear context for the rest of the paper and main focus of the paper.

Figure 1: the asterisk that marks the core location is hard to see when printed in black and white. Modify.

Line 202: what statistical basis was used to reject samples? What is the distinction between poor and not poorly preserved?

Line 280: cite (Eglinton and Hamilton, 1967) for odd over even predominance of n-alkanes.

Line 289: change sentence to ".…..(brGDGTs), produced by bacteria and that are abundant in soils, versus that.…..."

Line 290: add reference to (Sinninghe Damsté et al., 2002) for crenarchaeol

Figure 3: this figure is really cramped and it is very hard to see trends and what axis belongs to which curve.This figure should be split into several individual figures, I propose to have one focusing on the palynology, one on the inorganic and another on the organic geochemistry. Also, up-side-down axis and labels are hard to read.

Line 467: is there any other supporting information for the input of acidic peat input? For example, modern-day acidic peats are characterized by the dominance of the C31 ab-hopane (Dehmer, 1995; Pancost et al., 2002), which is normally only present in mature sediments. Is this seen in this immature marine sediment core as well? This could provide some key-supporting evidence for this statement of peat input.

Line 473-477: The authors should provide a ternary plot of the brGDGT distribution like used by (Sinninghe Damsté, 2016) and compare their brGDGT data with published

mineral soil (De Jonge et al., 2014) brGDGT distributions to rule out a significant non-terrestrial contribution.

Section 6.2: this section should be more extensive with a detailed discussion of the results from this sediment core in the context of previously published results and our understanding of Quaternary climate.

For the supplementary information, can the authors provide the abundances of the individual brGDGTs (and crenarchaeol) so that if the indices used for the soil-calibrations change in the future, the data can be easily recalculated and still be used in future studies.

David Naafs

References De Jonge, C., Hopmans, E.C., Zell, C.I., Kim, J.-H., Schouten, S., Sinninghe Damsté, J.S., 2014. Occurrence and abundance of 6-methyl branched glycerol dialkyl glycerol tetraethers in soils: implications for palaeoclimate reconstruction. Geochimica et Cosmochimica Acta 141, 97-112, doi: 10.1016/j.gca.2014.06.013.

Dehmer, J., 1995. Petrological and organic geochemical investigation of recent peats with known environments of deposition. International Journal of Coal Geology 28, 111-138, doi: 10.1016/0166-5162(95)00016-X.

Eglinton, G., Hamilton, R.J., 1967. Leaf Epicuticular Waxes. Science 156, 1322-1335, doi: 10.1126/science.156.3780.1322.

Haug, G.H., Ganopolski, A., Sigman, D.M., Rosell-Melé, A., Swann, G.E.A., Tiedemann, R., Jaccard, S.L., et al., 2005. North Pacific seasonality and the glaciation of North America 2.7 million years ago. Nature 433, 821-825, doi: 10.1038/nature03332.

Naafs, B.D.A., Hefter, J., Acton, G., Haug, G.H., Martínez-Garcia, A., Pancost, R., Stein, R., 2012. Strengthening of North American dust sources during the late Pliocene (2.7 Ma). Earth and Planetary Science Letters 317-318, 8-19, doi: 10.1016/j.epsl.2011.11.026.

Naafs, B.D.A., Hefter, J., Stein, R., 2013. Millennial-scale ice rafting events and Hudson Strait Heinrich(-like) Events during the late Pliocene and Pleistocene: a review. Quaternary Science Reviews 80, 1-28, doi: 10.1016/j.quascirev.2013.08.014.

Pancost, R.D., Baas, M., van Geel, B., Sinninghe Damsté, J.S., 2002. Biomarkers as proxies for plant inputs to peats: an example from a sub-boreal ombrotrophic bog. Organic Geochemistry 33, 675-690, doi: 10.1016/S0146-6380(02)00048-7.

Patterson, M.O., McKay, R., Naish, T., Escutia, C., Jimenez-Espejo, F.J., Raymo, M.E., Meyers, S.R., et al., 2014. Orbital forcing of the East Antarctic ice sheet during the Pliocene and Early Pleistocene. Nature Geoscience 7, 841, doi: 10.1038/ngeo2273.

Raymo, M.E., Lisiecki, L.E., Nisancioglu, K.H., 2006. Plio-Pleistocene Ice Volume, Antarctic Climate, and the Global $\delta$18O Record. Science 313, 492-495, doi: 10.1126/science.1123296.

Shakun, J.D., Raymo, M.E., Lea, D.W., 2016. An early Pleistocene Mg/Ca-$\delta$18O record from the Gulf of Mexico: Evaluating ice sheet size and pacing in the 41-kyr world. Paleoceanography 31, 1011-1027, doi: 10.1002/2016PA002956.

Sinninghe Damsté, J.S., 2016. Spatial heterogeneity of sources of branched tetraethers in shelf systems: The geochemistry of tetraethers in the Berau River delta (Kalimantan, Indonesia). Geochimica et Cosmochimica Acta 186, 13-31, doi: 10.1016/j.gca.2016.04.033.

Sinninghe Damsté, J.S., Schouten, S., Hopmans, E.C., van Duin, A.C.T., Geenevasen, J.A.J., 2002. Crenarchaeol: the characteristic core glycerol dibiphytanyl glycerol tetraether membrane lipid of cosmopolitan pelagic crenarchaeota. Journal of Lipid Research 43, 1641-1651, doi: 10.1194/jlr.M200148-JLR200.

---

## Referee Comment (RC3) · Anonymous Referee #3 · 12 Nov 2017

The authors use a multiproxy record from the southern North Sea to investigate phase relations between forcing and climate response in order to test the hypothesis that northern hemisphere obliquity forcing is the primary driver for glacial and interglacials during the Early Pleistocene. The paleoenvironmental record presented in this study certainly deserves publications as it contains important new information on paleoenvironmental change for a critical period in Earth history. The record also provides very interesting insights into how different terrestrial and marine proxies respond (or may not respond) to environmental changes.

However, after reading the manuscript, I am not entirely convinced that the selected

record is suitable to study leads and lags in high resolution and hence to test the main hypothesis of this study. The authors state that they "investigate the leads and lags of regional marine vs. terrestrial climatic cooling during MIS 102-92" (line 101). In fact, the record shows many gaps, and data resolution seems to be sufficient for high resolution studies from MIS 99 to MIS 94 only (Fig. 3). Unfortunately, the age model for this time interval (2.5-2.39 Ma), which is based on magnetostratigraphy and dinocyst events, does not seem to be very well constrained (Fig. 2).

Below is a figure with selected proxies for MIS 98-97, enlarged from Fig 3 and SI Fig. 2. The transition MIS98-97 have been chosen by the authors and described in detail to analyse forcings and responses in high resolution (line 442-445). The enlarged figures show that many proxies were measured at different depths and with gaps, which at least for some intervals hamper a robust identification of leads and lags. I also struggle to see the parallel initial decrease of cold water dinocysts and Sphagnum biomarkers (first two curves) and the final decrease in T/M ratio and d18O (last two curves), which, according to the authors, followed with a delay of a few thousand years. The arboreal pollen and T/M ratio curve shows large fluctuations and hardly reveal any clear trends. These fluctuations may have resulted from a) the extremely low pollen sum after exclusion of bisaccate pollen, and b) the fact that the pollen results were merged from two different sites. I have added the Pinus pollen percentages to demonstrate the large differences between the sample of the two sites (marked in red and blue). Given these limitations, I am not sure if the pollen curves are suitable at all for high resolution studies.

The manuscript would benefit from a longer and more detailed interpretation and discussion section and a critical evaluation of the limitations. I recommend the following revisions:

1. An excellent age control is critical for all high-resolution studies of leads and lags. The authors should therefore provide more information on how the specific section has been dated. Which tiepoints have been used? Fig. 2 is not sufficient and also includes

time intervals not relevant to this study.

2. It would be very helpful if the authors could provide a conceptual model describing in detail what they would expect to see in regard to the timing of each proxy, if obliquity forcing were the major driver.

3. The multiproxy approach makes the method chapter the longest section of the entire manuscript. Consider moving parts of the methods into the Supplementary Information and focus mainly on describing what the proxies show and discuss the methodological limitations relevant to this study.

4. The palaeoenvironmental interpretation of the record lacks depth and should be more detailed. In fact, the section "6.1 Paleoenvironmental settings" hardly contains any paleoenvironmental reconstructions and mainly focusses on the analysis of phase relations. For example, the first sentence does not explain why the pollen record indicates "generally warmer and more humid conditions". MIS 93 also contains surprisingly high Ericales percentages. Supplementary Fig. 1 is important to fully understand the paleoenvironmental settings and I recommend moving it into the main text.

5. The analysis of lead and lags needs to be more detailed in order to provide convincing evidence for the main conclusion. This requires the identification of multiple cycles with the same pattern (following the "conceptual model"), a precise dating of changes and, if possible, statistical analysis.

———————————————————

[Figure]

**Fig. 1.** Selected proxies described in line 442-445

---

## Author Comment (AC1) · 19 Dec 2017

Donders, T.H. et al. We thank the reviewers for their constructive and specific comments and will use them to improve the interpretation and data representation. Here we provide a first reply to the comments and indicate where we plan to make adjustments, and provide additional information to support our interpretations. We feel that with extension of the discussion and added detail as indicated below we are able to meet the concerns of all reviewers.

Reviewer Stijn de Schepper comment: Validity of age model: While the presented work is underpinned by previously published papers and insights into depositional environment (papers by Kuhlmann and co-authors), aspects of the age model can be questioned. The authors rely here on the G/M reversal and the X-event for constraining the age of their studied interval (L152–155). Kuhlman and Wong (2008) discuss in fact 4 different possible interpretations of the pmag. It seems very questionable to me that the very short-lived X-event (2.420–2.441 Ma, Cande and Kent, 1995) can be detected in the sedimentary record of a shallow sea by measuring the magnetic signal of discrete samples (Kuhlman and Wong, 2008). This event does not show up in u-channeled, high-resolution pmag records of the North Atlantic (e.g. Hoddell and Channell 2016; Channell et al 2016), neither has it been tied to the LR04 Marine Isotope Stratigraphy. The dinocyst bioevents generally point to the Plio-Pleistocene, but the events are not well-recognised (e.g. Barssidinium, M. choanophorum) or not calibrated (e.g. I. multiplexum) outside the North Sea Basin. This questions the age assigned to these events and thus also the age model. Using additional/different tiepoints that have been calibrated outside the North Sea Basin could provide more credibility to the age model used (see below). Based on these concerns about the age model, it remains uncertain 1) whether the cycles visible in the gamma-ray reflect the interval MIS102–96 and 2) whether these are truly, consecutive (i.e. no erosional events in between) G-IG cycles.

Reply: The comments regarding the age model focus on three aspects; the validity of the paleomagnetic signal and, consequently, the completeness and correct assignment of the stratigraphy at A15-3/4 to MIS 102-96, and the use of the dinocyst biozonation. Firstly, based on the combined stratigraphic and detailed 3D seismic interpretations and overall fine grained (clays to silts) deposits all point to a continuously aggrading system in the interval we report. There is evidence of small hiatuses above (first around 2.1 Ma) and significant hiatuses below (intervals within the Early Pliocene and Miocene, particularly the Mid Miocene Unconformity) the selected interval, which is why we excluded these intervals in this publication. Indeed, in the excluded intervals erosional surfaces (beside the obvious MMU) are well recognizable in the seismic property data (Kuhlmann and Wong, 2008), where the high-resolution 3D volume resolves e.g. (iceberg) scour marks and truncated clinoforms. The seismic data thus serve as

an important control on our stratigraphic interpretation. In the intervals with erosive signals, the associated palynological signals point to much more shallow and near terrestrial conditions that are typically associated with erosive conditions (Kuhlmann et al., 2006a)

For the reported MIS102-96 interval, the typical cyclic pattern of the gamma ray is traceable across several wells in the central part of the entire southern North Sea (see Kuhlman et al. 2006ab as well as in the seismic interpretations presented in our supplementary data). Crucially, the Pmag has been measured first by a continuous paleomagnetic downhole logging tool, Geological High-resolution Magnetic Tool (GHMT) by Schlumberger, in wells A15-3 and B16-1 (see description in Kuhlmann et al., 2006a), which is a rarely available tool and therefore an important addition to the interpretation. This continuous signal is present in two wells in the same log zone and has subsequently been verified by discrete samples taken from continuous cores in well A15-3 (Kuhlmann et al., 2006a), and the interpretation relies on the combined signal from borehole logging and core measurements. Secondly, owing to the coastal proximity, the thickness of the North Sea succession and therewith sedimentation rates of the investigated interval is far higher than any North Atlantic site, which greatly increases the chance of recovery of the X-event. Our approximately 250 kyr record is represented by an sediment thickness from over 160 m of fine-grained sediment. While the independent position of the X-event is not included in i.e. the LR stack, there is additional recent evidence that supports our interpretation. Noorbergen et al. (2015) has carried out a detailed study of a land-based section (Noordwijk well) that represents approximately the same interval as our 15-3/4 study. The Noordwijk record contains both palynology and detailed stable isotope stratigraphy, and it includes a direct correlation with the A15-3 well, including the quantitative abundance signals on palynology. At this site, carbonate preservation was much better and more sample material was available, providing a much more complete benthic isotope record. Based on the Noordwijk data, Noorbergen et al (2015) established a tuning to LR04, which is valid for A15-3/4 as well. The 4 options for paleomagnetic interpretation in Kuhlman and Wong (2008) pointed at

by the reviewer, are already presented in Kuhlmann et al. (2006a), and represent the theoretical ties when only Pmag data would be considered. The key to our record is an integrated Pmag, isotope stratigraphic, seismic stratigraphic and palynological biozonation that exclude the other options and all converge on the present interpretation. We recognize that the evidence from the Noordwijk well (Noorbergen et al, 2015) was insufficiently represented in our manuscript and we will incorporate this study in our discussion to strengthen our interpretation, and we will refer to the available evidence on hiatuses.

The bioevents in the North Sea basin, specifically the acmes, indeed have a clear regional character, but within the basin allow a high resolution well correlation (Kuhlmann et al., 2006b). While the age model and bioevents have been discussed in Kuhlmann et al. (2006a) and are used for this publication, their validity is significantly strengthened by the tuning approach of Noorbergen et al. (2015). That paper describes the occurrence of I. multiplexum in both the A15-3 well and Noordwijk well, which has been tied to an acme in MIS 97/98 in this basin. Based on the comments, we have reviewed the dinocyst events and the suggested inclusion of the additional markers strengthens our interpretation. In the revision we will provide an updated table with the age of the events used and update the age-depth model where needed. This revision is not expected to alter the interpretations of the MIS102-92 interval.

Comments reviewer: The leads/lags between climate proxies and sea level are not so clearly visible as the authors claim in the abstract and conclusions. The leads/lags are not clearly demonstrated on a figure, or more importantly using statistical techniques.

Reply: Lead –lags signals that we infer are based on the G-IC cycle (MIS 98-97-96) in our record that is best resolved in all available proxies. A statistical approach would require multiple of these successions with similar sampling resolution which, unfortunately, is not available. The stratigraphic record is not fully cored, but only in part (see fig 2) and part of the proxies (palynology and organic geochemistry) supplemented by side wall cores. The strength and value of the record is in the expanded nature and

good reflection of both marine and terrestrial signals, which is a rare occasion. Based on the available evidence we infer a lead-lag relation of (crucially) signals that are all coming from the same source material. While the overall climate signal between land, sea surface and sea level is indeed in phase ("vary in concert"), there are small lead –lags relations in the data that we point to, we will highlight these different aspect more extensively in the revised manuscript in such a way that it is clear that no exact duration of the phase lag has been inferred yet (this would require a more continuous record).

Comment reviewer: What is the effect of using cutting samples (caving, reworking) on your interpretations?

Reply: The effect of the cuttings is very minimal as the majority of the samples is from cores or side wall cores (which are intact samples obtained after drilling). The cuttings are used here to increase resolution of the palynological samples, and are based on larger chips that have been cleaned before treatment. Importantly, no PDC (power drill bit) has been used so the cutting material has not been ground into a fine paste as is a common practice in many recent wells. The expanded sediment package helps limit the caving problems as the time resolution is high. We will provide a table with exact sample type and proxy, but we can state that all organic and carbonate proxies have been measured on core or sidewall cores and the key conclusions are not depending on cutting material. Other wells in the region that have been studied (by TNO Geological Survey of the Netherlands), internal reports) on cutting material could be correlated confidently to A15-3/4, and independently verified by 3D seismic interpretations.

Comment reviewer: Environmental signal from dinocysts; The 4 species used to indicate a warm water signal are all coastal, shallow water species

Reply: The reviewer is absolutely right that the 4 selected dinocyst taxa indicate coastal conditions. In fact this is our main purpose for displaying them, and conclusions based on their abundance refer to their indication of coastal conditions. Our climatic interpretations rely on the cool water taxa as we tried to optimally separate the in-/offshore
and climatic trends. The confusion is in our use of the phrase " … indicate generally
warm, coastal waters", while we principally use them for the latter. We shall explain this
point better in the revision. The principal source of the terrestrial palynomorphs is from
the Eridanos paleoriver (as verified in a source area study by Kuhlmann et al., 2004).
The detailed seismic interpretation provides further important control on the direction
of river progradation. The component most sensitive to the T/M index related to differ-
ential transport processes, the bisaccate pollen, are here tested for their effect on the
ratio by including and excluding them (Fig. 1). The resulting ratio with bisaccate pollen
excluded is slightly lower, but the relation between both ratios is very strong and hence
no indications for phases of differential transport are present. This additional figure will
be included in the supplementary data.

Minor points by reviewer S. de Schepper

All suggested textual comments will be clarified and/or adjusted

The suggested use of the Gibbard and Lewin 2016 paleogeographic reconstruction will
be considered, it will not alter the implications of our study

Comment L121 "different water types": water masses in Fig 1 refer mostly to the origin
of the fresh water inflows

Comment L126–128 Please provide a timeframe Reply: the Eridanos delta was ac-
tive during most of the Neogene and early Pleistocene, and progressively prograded
towards the study site

Comment L157–162 comment on the depositional model;This model would get more
credibility if this has also been demonstrated for late Pleistocene glacial/interglacial
cycles. Would the SL drop of up to 60 m in these glacials (e.g. Miller et al. 2005;
Bintanja et al. 2005) not provide a stronger control on the sedimentation (rather than
hydrography)? Reply: the relation between grain size and G-IG cycles is regionally only

and valid as long as the site is permanently marine. Available foraminifera and seismic data indicate water depths of 300-100 m in the reported interval (Kuhlmann et al., 2006a; Huuse et al., 2001). In later glacial stages as the Eridanos system is abandoned and more extensive glaciations cover the Scandinavian shield and the Southern North Sea basin is either dry or very shallow this depositional system proposed by Kuhlmann and Wong (2008) is no longer valid. Also the study by Noorbergen et al (2015) on the Noordwijk well confirms that " . . . the finer grained intervals coincide with d18O maxima implying increased ice sheet volume and lowered eustatic sea levels."

Comment L165 How was the age model transferred to LR04 MIS? Reply: GR breaks were picked as inflection points of the LR04 MIS transitions; this will be clarified in the revision in the tie point table.

Comment L202 How was recrystallization and dissolution determined? Reply: Preservation was based on a visual inspection and assignment of a relative scale of 1-5 of preservation, after which the poorest 2 classes were discarded. The best preserved specimens (cat. 1) had shiny tests (original wall calcite) and showed no signs of overgrowth. Category 2 specimens showed signs of overgrowth but were not recrystallized and cat. 3 specimens were dull and overgrown by a thin layer of secondary calcite. Cat 4-5 specimens were discarded because primary calcite was (nearly) absent. While we are aware of the importance of SEM work for detailed preservational assessments the aim was to establish the phase relation with the GR cycles.

Comment L273 Why were there only relative abundances calculated? Reply: No lycopodium counts were available, needed for calculations of concentrations, due to part industry origin of the datasets (data produced by authors for industry purpose).

Comment: L359 not convincing Reply:reviewer referring to phrase 'The Cassidulina teretis $\delta$18O ($\delta$18Ob) confirms the relation between glacial stages and fine grained sediment as proposed by Kuhlman et al. (2006a,b)": Apart from our data, the benthic ($\delta$18Ob) from the nearby Noordwijk well (Noorbergen et al., 2015) now independently

confirms the relation between glacial stages and fine grained sediment as proposed by Kuhlman et al. (2006a).

Comment L383 Which fresh water algae did you find? Reply: Pediastrum and Botryococcus (see supplementary data)

Comment: L401-402 What does the n-C23 Sphagnum biomarker indicate? Reply: development of boreal (moist/cool) climate and influx

Comment Chapter 6.1 is confusing. reply: we will revise the text for inconsistencies

Comment L434 Effect of SL on pollen is addressed here, but the effect of SL on the dinocyst record is not discussed in the MS. Reply: the coastal dinocyst index is especially included to document the combined influence of coastal progradation and sea level change. Due to the earlier confusion on the use as warm water indicators, this point was perhaps overseen by the reviewer.

Comment L485, L535 Onset/intensification have been used intermixed. Reply: valid point that we will adjust

Comment: L510 Speculation Reply: yes, but consistent with the effect of obliquity forcing

Comment L518 Severe cooling. Subjective comment. Reply: the cooling is relative to late Pliocene conditions and in that respect severe. We will specify more exactly the degree of cooling based on the brGDGT data that, although not always in phase with the other proxies, does give a temperature range for the G-IG cycles.

Comment: Please provide a list with the tiepoints. Reply: we will add this together with the MIS transitions, but all age tie points are reported in Kuhlmann et al, 2006ab

[Figure]

**Fig. 1.** Figure 1: terrestrial / marine palynomorph ratios with in- and exclusion of bisaccate pollen.

---

## Author Comment (AC2) · 19 Dec 2017

We thank the reviewers for their constructive and specific comments and will use them to improve the interpretation and data representation. Here we provide a first reply to the comments and indicate where we plan to make adjustments, and provide additional information to support our interpretations. We feel that with extension of the discussion and added detail as indicated below we are able to meet the concerns of all reviewers.

Reviewer David Naafs Comment: Discussion on the phase relationship "the discussion

[Figure]

on this specific topic in this manuscript is rather limited and is missing a discussion of crucial prior work on this topic" Reply : We originally aimed at providing a compact paper focusing on the evidence on phase relations we can provide from the new data, but we acknowledge that more information is available. We will expand the introduction and discussion on this matter following the suggestions of the reviewer to provide a more balanced assessment of the forcing mechanisms.

Comment: In addition, I wonder whether the age model is robust enough. The low-resolution benthic d18O record of this site does not always look like the LR04 stack. Reply: the validity of the age model is addressed in detail in the replies to Stijn de Schepper, which we will not repeat here, but in short we will focus on more extensive discussion of records in the same basin, in particular the Noordwijk record from Noor-bergen et al. (2015) where an independent tuning to LR04 is available. As the reviewer acknowledges, the key results of our paper however, depend on the internal relations between the proxies from the same record and do not rely on an exact match with the LR04 stack.

Minor comments:

Comment Line 51-55: this is a bit of a weird ending of the abstract, especially in the context of the main focus of the paper that is stated at the beginning of the abstract. The authors should end the abstract with a clear conclusion of what, according to their work, the phase relation is between forcing and climatic response. Reply: the abstract will be adapted to better reflect the conclusion

Comment Line 66: a full review paper on IRD in the North Atlantic during the Plio/Pleistocene is given in (Naafs et al., 2013). Reply: add IRD references is a good suggestion

Comment Line 73-78: somewhere make reference to mechanism proposed. Reply: will add reference to proposed forcing mechanism (Haug et al., 2005)

[Figure]

Comment Line 82-88: here other recent publications that refute or support Raymo's hypothesis. Reply: we will add more extensive discussion on the phase relationship between forcing and climatic response in the early Pleistocene

Comment Line 202: what statistical basis was used to reject samples? What is the distinction between poor and not poorly preserved? Reply: Preservation was based on a visual inspection and assignment of a relative scale of 1-5 of preservation, after which the poorest 2 classes were discarded. The best preserved specimens (cat. 1) had shiny tests (original wall calcite) and showed no signs of overgrowth. Category 2 specimens showed signs of overgrowth but were not recrystallized and cat. 3 specimens were dull and overgrown by a thin layer of secondary calcite. Cat 4-5 specimens were discarded because primary calcite was (nearly) absent.

Comment Line 280: cite (Eglinton and Hamilton, 1967) for odd over even predominance of nalkanes. Reply: we will cite Eglinton and Hamilton, 1967 on n-alkanes

Comment Line 289 change sentence to "brGDGTs), produced by bacteria and that are abundant in soils, versus that:.." Reply: textual comments will adopted

Comment Line 290: add reference Reply: we will cite Sinninghe Damsté et al., 2002 for crenarchaeol

Comment Line 467: is there any other supporting information for the input of acidic peat input? For example, modern-day acidic peats are characterized by the dominance of the C31ab-hopane (Dehmer, 1995; Pancost et al., 2002), which is normally only present inmature sediments. Reply: as seen in the expanded pollen diagram (Fig. S2), Sphagnum spores are also mostly enhanced in the glacial MIS intervals in support of the C23 biomarker. In the revision we plan to also include the isomers index of the de C31 hopanes that when immature, provide an indication for acidic peat

Comment Line 473-477 The authors should provide a ternary plot of the brGDGT distribution to rule out a significant non-terrestrial contribution; Reply: construction of a

ternary diagram is in progress and will be added in the revision. Comment Fig 3 readability

Reply: the aim of the figure is to compare various proxies and they therefore need to be together. The figure is now rotated but will be horizontal in final version, improving visibility

Comment Suppl data Reply: We will add the absolute abundances of the individual brGDGTs (and crenarchaeol) to enable recalculations

---

## Author Comment (AC3) · 19 Dec 2017

Donders, T.H. et al. We thank the reviewers for their constructive and specific comments and will use them to improve the interpretation and data representation. Here we provide a first reply to the comments and indicate where we plan to make adjustments, and provide additional information to support our interpretations. We feel that with extension of the discussion and added detail as indicated below we are able to meet the concerns of all reviewers.

Anonymous Referee #3

[Figure]

Comment: The arboreal pollen and T/M ratio curve shows large fluctuations and hardly reveal any clear trends. These fluctuations may have resulted from a) the extremely low pollen sum after exclusion of bisaccate pollen, and b) the fact that the pollen results were merged from two different sites.

Reply: as Rev. #3 suggests, the AP curve shows variability. It is however clear that the glacial intervals AP values do not exceed 20%, except for one sample, and are consistently associated with increased Ericaceae. Interglacial AP values are clearly enhanced between 20 and 50 %, so we strongly disagree with the statement that there is no clear trend. The detailed pollen diagram in the supplementary data shows consistent abundance changes for the combined (spliced) dataset for e.g. Ericaceae, ferns, Picea. The variability in the Pinus curve is also visible in the sections that come from a single core, e.g. in MIS 95 and thus not a product of the splice. The splice is based on the high resolution GR record (verified by the dinocyst events), which provides a total of 15 tie points that produced a completely linear well tie (see Fig. 1).

Comment 1:An excellent age control is critical for all high-resolution studies of leads and lags. The authors should therefore provide more information on how the specific section has been dated.

Reply: see validation by Noorbergen et al (2015) study outlined in the reply to S. de Schepper, including tie points and age model construction. In short, the basic age model is outlined in Kuhlmann et al., 2006a, we have only transferred this age dating on an age scale. The leads and lags infer changes between proxies from the same record and in that sense are independent of an exact age model, although obviously desirable. Tie points and a review of the ages used in Kuhlmann et al., 2006a will be provided in the revision and is not expected to alter the outcomes.

Comments 3&4: The multiproxy approach makes the method chapter the longest section of the entire manuscript. Consider moving parts of the methods into the Supplementary Information and focus mainly on describing what the proxies show and

discuss the methodological limitations relevant to this study. The palaeoenvironmental interpretation of the record lacks depth and should be more detailed.

Reply: we do provide the outline of the interpretations for each proxy in the methods section of the original manuscript. We will expand the palaeoenvironmental interpretation, particularly on the pollen data, referring in more detail to the full record in the supplement. Based on this revision we will decide on moving sections to the supplement as is suggested by Rev. #3. and highlight individual results more explicitly where needed.

Comments 2&5: It would be very helpful if the authors could provide a conceptual model describing in detail what they would expect to see in regard to the timing of each proxy, if obliquity forcing were the major driver. The analysis of lead and lags needs to be more detailed in order to provide convincing evidence for the main conclusion. I also struggle to see the parallel initial decrease of cold water dinocysts and Sphagnum biomarkers (first two curves) and the final decrease in T/M ratio and d18O (last two curves), which, according to the authors, followed with a delay of a few thousand years.

Reply: Statistical analysis of the lead-lag relations is desirable but unfortunately not possible due to the limits of the record recovery, hence we choose to focus on the best resolved and completely cored G-IG cycle (MIS 98-97-96). We will improve the description of our specific observations on the leads and lags and link them, as suggested by Rev #3, to our forcing scenario in relation to the expanded discussion on this topic (as was suggested by reviewer D. Naafs). In particular, as questioned by Rev#3; Decreases in cold water dinocysts and Sphagnum biomarkers (first two curves) and the final decrease in T/M ratio and d18O are based on high values of the first two in the early half of MIS 98 (shaded interval), after which the T/M increases only in the second half of MIS98 (and correlated LR04 d18O signal, but this detail depends on uncertainty in the age model). The key curve to assess are the cold water dinocysts and not the coastal signal, which is probably causing the remarks by rev #3.

Cited references (used in replies to all reviewers)

Haug, G.H., Ganopolski, A., Sigman, D.M., Rosell-Melé, A., Swann, G.E.A., Tiedemann, R., Jaccard, S.L., et al., 2005. North Pacific seasonality and the glaciation of North America 2.7 million years ago. Nature 433, 821-825, doi: 10.1038/nature03332.

Eglinton, G., Hamilton, R.J., 1967. Leaf Epicuticular Waxes. Science 156, 1322-1335, doi: 10.1126/science.156.3780.1322.

Huuse, M., Lykke-Andersen, H., Michelsen, O., 2001. Cenozoic evolution of the eastern North Sea Basin — new evidence from high-resolution and conventional seismic data. Marine Geology 177: 243–269.

Kuhlmann, G. & Wong, T.E., 2008. Pliocene paleoenvironment evolution as interpreted from 3D-seismic data in the southern North Sea, Dutch offshore sector. Marine and Petroleum Geology 25: 173-189.

Kuhlmann, G., Pedersen, R.-B., de Boer, P., Wong, Th.E., 2004. Provenance of Pliocene sediments and paleoenvironmental change in the southern North Sea region using Sm/Nd (samarium-neodymium) provenance ages and clay mineralogy: Sedimentary Geology 171: 205-226.

Kuhlmann, G., Langereis, C.G., Munsterman, D., van Leeuwen, R.-J., Verreussel, R., Meulenkamp, J., Wong, Th.E., 2006a. Chronostratigraphy of Late Neogene sediments in the southern North Sea Basin and paleoenvironmental interpretations. Palaeogeography, Palaeoclimatology, Palaeoecology 239: 426–455.

Kuhlmann, G., Langereis, C.G., Munsterman, D., van Leeuwen, R.-J., Verreussel, R., Meulenkamp, J.E., Wong, Th.E., 2006b. Integrated chronostratigraphy of the Pliocene–Pleistocene interval and its relation to the regional stratigraphical stages in the southern North Sea region. Netherlands Journal of Geosciences - Geologie en Mijnbouw 85 (1): 19–35.

Naafs, B.D.A., Hefter, J., Stein, R., 2013. Millennial-scale ice rafting events and Hudson Strait Heinrich(-like) Events during the late Pliocene and Pleistocene: a review. Quaternary Science Reviews 80, 1-28, doi: 10.1016/j.quascirev.2013.08.014.

Noorbergen, L. J.; Lourens, L. J.; Munsterman, D. K.; Verreussel, R.M.C.H., 2015. Stable isotope stratigraphy of the early Quaternary of borehole Noordwijk, southern North Sea .Quaternary International, volume 386, pp. 148 - 157

Sinninghe Damsté, J.S., Schouten, S., Hopmans, E.C., van Duin, A.C.T., Geenevasen, J.A.J., 2002. Crenarchaeol: the characteristic core glycerol dibiphytanyl glycerol tetraether membrane lipid of cosmopolitan pelagic crenarchaeota. Journal of Lipid Research 43, 1641-1651, doi: 10.1194/jlr.M200148-JLR200.
* * *
**GR correlation points**

y = 1.039x - 22.219
R² = 0.9981

◆ GR correlation points

— Linear (GR correlation points)

Depth A15-04 (mbsl)

Depth A15-03 (mbsl)

**Fig. 1.** Figure 1: Well tie correlation points indicate a clear linear relation between the wells A15-3 and A15-4

---

## Author Response (AR1)

**Reply and changes made to reviewer comments on 'Land–sea coupling of Early Pleistocene glacial cycles in the southern North Sea exhibit dominant Northern Hemisphere forcing'**

**Donders, T.H. et al.**

We thank the reviewers for their constructive and specific comments and have used them to improve the interpretation and data representation. Here we provide a full reply to the comments and indicate where we have made adjustments, and provide additional information to support our interpretations. We have checked references in detail and added doi numbers. We feel that with extension of the discussion and added detail as indicated below we are able to meet the concerns of all reviewers.

**Reviewer #1: Stijn de Schepper**

Comment on validity of age model: While the presented work is underpinned by previously published papers and insights into depositional environment (papers by Kuhlmann and co-authors), aspects of the age model can be questioned. The authors rely here on the G/M reversal and the X-event for constraining the age of their studied interval (L152–155). Kuhlman and Wong (2008) discuss in fact 4 different possible interpretations of the pmag. It seems very questionable to me that the very shortlived X-event (2.420–2.441 Ma, Cande and Kent, 1995) can be detected in the sedimentary record of a shallow sea by measuring the magnetic signal of discrete samples (Kuhlman and Wong, 2008). This event does not show up in u-channeled, high-resolution pmag records of the North Atlantic (e.g. Hoddell and Channell 2016; Channell et al 2016), neither has it been tied to the LR04 Marine Isotope Stratigraphy. The dinocyst bioevents generally point to the Plio-Pleistocene, but the events are not well-recognised (e.g. Barssidinium, M. choanophorum) or not calibrated (e.g. I. multiplexum) outside the North Sea Basin. This questions the age assigned to these events and thus also the age model. Using additional/different tiepoints that have been calibrated outside the North Sea Basin could provide more credibility to the age model used (see below). Based on these concerns about the age model, it remains uncertain 1) whether the cycles visible in the gamma-ray reflect the interval MIS102–96 and 2) whether these are truly, consecutive (i.e. no erosional events in between) G-IG cycles.

Reply: The comments of the reviewers regarding the age model focus on three aspects;

- 1. the sedimentary setting and validity of the paleomagnetic signal and, consequently,
- 2. the completeness and correct assignment of the stratigraphy at A15-3/4 to MIS 102-96, and
- 3. the use of the dinocyst biozonation.

1: Firstly, based on the combined stratigraphic and detailed 3D seismic interpretations and overall fine grained (clays to silts) deposits all point to a continuously aggrading system in the interval we report. There is evidence of small hiatuses above (first around 2.1 Ma) and significant hiatuses below (intervals within the Early Pliocene and Miocene, particularly the Mid Miocene Unconformity) the selected interval, which is why we excluded these intervals in this publication. Indeed, in the excluded intervals erosional surfaces (beside the obvious MMU) are well recognizable in the seismic property data (Kuhlmann and Wong, 2008), where the high-resolution 3D volume resolves e.g. (iceberg) scour marks

and truncated clinoforms. The seismic data thus serve as an important control on our stratigraphic interpretation. In the intervals with erosive signals, the associated palynological signals point to much more shallow and near terrestrial conditions that are typically associated with erosive conditions (Kuhlmann et al., 2006a).

For the reported MIS102-96 interval, the typical cyclic pattern of the gamma ray is traceable across several wells in the central part of the entire southern North Sea (see Kuhlman et al. 2006ab as well as in the seismic interpretations presented in our supplementary data). Crucially, the Pmag has been measured first by a continuous paleomagnetic downhole logging tool, Geological High-resolution Magnetic Tool (GHMT) by Schlumberger, in wells A15-3 and B16-1 (see description in Kuhlmann et al., 2006a), which is a rarely available tool and therefore an important addition to the interpretation. This continuous signal is present in **two** wells in the same log zone and has subsequently been verified by discrete samples taken from continuous cores in well A15-3 (Kuhlmann et al., 2006a), and the interpretation relies on the combined signal from borehole logging and core measurements. Secondly, owing to the coastal proximity, the thickness of the North Sea succession and therewith sedimentation rates of the investigated interval is far higher than any North Atlantic site, which greatly increases the chance of recovery of the X-event. Our approximately 250 kyr record is represented by a sediment thickness of over 160 m of fine-grained sediment.

2: While the independent position of the X-event is not included in i.e. the LR stack, there is additional recent evidence that supports our interpretation. Noorbergen et al. (2015) has carried out a detailed study of a land-based section (Noordwijk well) that represents approximately the same interval as our 15-3/4 study. The Noordwijk record contains both palynology and detailed stable isotope stratigraphy, and it includes a direct correlation with the A15-3 well, including the quantitative abundance signals on palynology. At this site, carbonate preservation was much better and more sample material was available, providing a much more complete benthic isotope record. Based on the Noordwijk data, Noorbergen et al (2015) established a tuning to LR04, which is valid for A15-3/4 as well. The 4 options for paleomagnetic interpretation in Kuhlman and Wong (2008) pointed at by the reviewer, are already presented in Kuhlmann et al. (2006a), and represent the theoretical ties when only Pmag data would be considered. The key to our record is an integrated Pmag, isotope stratigraphic, seismic stratigraphic and palynological biozonation that exclude the other options and all converge on the present interpretation as presented in Table S1 and Figure S2. We recognize that the evidence from the Noordwijk well (Noorbergen et al, 2015) was insufficiently represented in our manuscript and we have incorporated this study in section 3 and the discussion to strengthen our interpretation, and we refer to the available evidence on hiatuses. Also, we recap shortly the available age dating information (as outlined here).

3. The bioevents in the North Sea basin, specifically the acmes, indeed have a clear regional character, but within the basin allow a high resolution well correlation (Kuhlmann et al., 2006b). While the age model and bioevents have been discussed in Kuhlmann et al. (2006a) and are used for this publication, their validity is significantly strengthened by the tuning approach of Noorbergen et al. (2015). That paper describes the occurrence of *I. multiplexum* in both the A15-3 well and Noordwijk well, which has been tied to an acme in MIS 97/98 in this basin. Based on the comments, we have reviewed the dinocyst

events and the suggested inclusion of the additional markers strengthens our interpretation. In the revision, we provide a new Table S1 with the age and sources of the bioevents used, and have updated the age-depth model where needed to included uncertainties explicitly. As expected, the revision did not alter the age interpretations of the MIS102-92 interval.

**Comment: The leads/lags between climate proxies and sea level are not so clearly visible as the authors claim in the abstract and conclusions. The leads/lags are not clearly demonstrated on a figure, or more importantly using statistical techniques.**

Reply: Lead –lags signals that we infer are mainly based on the G-IC cycle (MIS 98-97-96) in our record that is best resolved in all available proxies. A statistical approach would require multiple of these successions with similar sampling resolution which, unfortunately, is not available. The stratigraphic record is not fully cored, but only in part (see fig 2) and part of the proxies (palynology and organic geochemistry) supplemented by side wall cores. The strength and value of the record is in the expanded nature and good reflection of both marine and terrestrial signals, which is a rare occasion. Based on the available evidence we infer a lead-lag relation of (crucially) signals that are all coming from the same source material. While the overall climate signal between land, sea surface and sea level is indeed in phase ("vary in concert"), there are small lead –lags relations in the data that we point to. The best resolved and across alonger portion of the record is that of the AP% and T/M ratio that, crucially, are based on the same palynological analysis. Based on the available records, AP% declines with a lead of between 3-8 kyr relative to the T/M increases based on the present age model. While we have confidence in our age model, the exact duration of the glacial vs interglacial part of each sedimentary cycle is not constrained, only the transitions. For this reason we want to refrain from providing too exact numbers on the lead-lag relations, but have added further discussion in 6.1 on the AP% and T/M leadlags and indicated the offsets in Fig S2.

**Comment: What is the effect of using cutting samples (caving, reworking) on your interpretations?**

Reply: The effect of the cuttings is very minimal as the majority of the samples is from cores or side wall cores (which are intact samples obtained after drilling). The cuttings are used here to increase resolution of the palynological samples, and are based on larger chips that have been cleaned before treatment. Importantly, no PDC (power drill bit) has been used so the cutting material has not been ground into a fine paste as is a common practice in many recent wells. The expanded sediment package helps limit the caving problems as the time resolution is high. We will provide a table with exact sample type and proxy, but we can state that all organic and carbonate proxies have been measured on core or sidewall cores and the key conclusions are not depending on cutting material. Other wells in the region that have been studied (by TNO Geological Survey of the Netherlands), internal reports) on cutting material could be correlated confidently to A15-3/4, and independently verified by 3D seismic interpretations. We added some explanation on the use of the cutting material.

**Comment: Environmental signal from dinocysts; The 4 species used to indicate a warm water signal are all coastal, shallow water species (L225–227). Their distribution in the shallow North Sea Basin**

could be strongly affected by SL fluctuations at the beginning of the Early Pliocene. Versteegh (1994) therefore does not include these taxa in a warm-cool index. Furthermore, L. machaerophorum is often used to indicate river input and sea level fluctuations (Holzwarth et al. 2010). How do you disentangle the effect of sea level and temperature for these 4 species, when their distribution could be affected by both? The T/M ratio is interpreted as a relative SL indicator. While this intuitively seems correct, I wonder if the relation is that simple? Terrestrial palynomorphs are affected by transport patterns (wind, position of rivers) and could thereby influence the sea level interpretation?

Reply: The reviewer is absolutely right that the 4 selected dinocyst taxa indicate largely coastal conditions. In fact this is our main purpose for displaying them, and conclusions based on their abundance refer to their indication of coastal conditions. Our climatic interpretations regarding dincysts rely on solely the cool water taxa as we tried to optimally separate the in-/offshore and climatic trends. The confusion is in our use of the phrase " ... *indicate generally warm, coastal waters*", while we principally use them for the latter. We have explained this point better in the revision. On the second point regarding the T/M ratio: The principal source of the terrestrial palynomorphs is from the Eridanos paleoriver (as verified in a source area study by Kuhlmann et al., 2004). The detailed seismic interpretation provides further important control on the direction of river progradation. The component most sensitive to the T/M index related to differential transport processes, the bisaccate pollen, are here tested for their effect on the ratio by including and excluding them (Fig. 1). The resulting ratio with bisaccate pollen excluded is slightly lower, but the relation between both ratios is very strong and hence no indications for phases of differential transport are present that impact the T/M ratio. This explanation was added to the discussion and the additional figure was included in the supplementary data (Fig. S5).

*Figure 1: terrestrial / marine palynomorph ratios with in- and exclusion of bisaccate pollen.*

**Minor points**

**L45 There is hardly proxy data to say something about MIS 100** Reply: 100 excluded and 94 was included in the summary

**L47 Freshwater flux is not really supported by the fresh water algae.** Reply: statement changed to the record of stratification from *L. machaerophorum*, better reflecting the discussion

**L50-51 Confusing. Please rephrase.** Reply: rephrased to "The record provides evidence for a dominantly NH driven cooling that leads the glacial build up and varies on obliquity timescale."

L52 SST is not a good indicator of migration of a watermass. Microfossil assemblage could help you with identifying such migration, but not SST alone. Reply: rephrased, the SST signal from the microfossil assemblage was indeed meant. "indicated by cool-water microfossil assemblages"

L73 space missing before "and" Reply: corrected

**L105 rephrase "but which stratigraphic position"**

Reply: changed to "although its stratigraphic position and original definition are not well defined"

L110 During the Neogene, there could have been a southerly connection between the North Sea and Atlantic (see reconstructions of e.g. Gibbard and Lewin 2003, 2016). It might be worth to use the more recent Gibbard and Lewin 2016 palaeogeographic reconstruction instead of Zeigler 1990 (Figure 1). Reply: The suggested use of the Gibbard and Lewin 2016 paleogeographic reconstruction was considered but to display the center of deposition we prefer to use the current geographic boundaries with the current sedimentary infill and overlay of paleoflows, and refer to the paleogeographic reconstructions for more detail.

L121 "different water types": water masses in Fig 1 refer mostly to the origin of the fresh water inflows

Reply: caption Fig. 1 corrected

**L126–128 Please provide a timeframe:**

Reply: changed to the Eridanos delta was active during most of the Neogene and Early Pleistocene, and progressively prograded towards the study site

L157–162 This (depositional) model would get more credibility if this has also been demonstrated for late Pleistocene glacial/interglacial cycles. Would the SL drop of up to 60 m in these glacials (e.g. Miller

**et al. 2005; Bintanja et al. 2005) not provide a stronger control on the sedimentation (rather than hydrography)?**

Reply: the relation between grain size and G-IG cycles is regionally only and valid as long as the site is permanently marine. Available foraminifera and seismic data indicate water depths of 300-100 m in the reported interval (Kuhlmann et al., 2006a; Huuse et al., 2001). In later glacial stages as the Eridanos system is abandoned and more extensive glaciations cover the Scandinavian shield and the Southern North Sea basin is either dry or very shallow this depositional system proposed by Kuhlmann and Wong (2008) is no longer valid. Also the study by Noorbergen et al (2015) on the Noordwijk well confirms that " *... the finer grained intervals coincide with d180 maxima implying increased ice sheet volume and lowered eustatic sea levels.*" Reference to the Noordwijk study has been included

**L165 How was the age model transferred to LR04 MIS?**

Reply: GR breaks were picked as inflection points of the LR04 MIS transitions (allowing for a 20 kyr uncertainty around individual ties) and interpolated through a smoothing spline; this is now clarified clearly in section 3 and Table S1.

**L174–175 Please check also De Schepper et al. 2017.**

Reply: added in table S1

**L177–L186 Does this paragraph belong in the age model section?**

Reply: we think it fits best here as it describes the regional setting and the seismic interpretations provide basin-wide correlations.

**L190 C. teretis in italics.**

**Reply: corrected**

**L202 How was recrystallization and dissolution determined?**

Reply: Preservation was based on a visual inspection and assignment of a relative scale of 1-5 of preservation, after which the poorest 2 classes were discarded. The best preserved specimens (cat. 1) had shiny tests (original wall calcite) and showed no signs of overgrowth. Category 2 specimens showed signs of overgrowth but were not recrystallized and cat. 3 specimens were dull and overgrown by a thin layer of secondary calcite. Cat 4-5 specimens were discarded because primary calcite was (nearly) absent. While we are aware of the importance of SEM work for detailed preservational assessments the aim was to establish the phase relation with the GR cycles. These details have been added to the methods.

**L211 de Vernal (no capital D).** Reply: corrected

L270 delete ", dinocysts":

Reply: we deleted "dinoflagellate cyst" as the term dinocyst had already been introduced earlier in the text.

**L273 Why were there only relative abundances calculated?**

Reply: No *Lycopodium* marker counts were available, needed for calculations of concentrations, due to part industry origin of the datasets.

**L304 Delete "For TOC determination".**

Reply: corrected

L359 not convincing (reviewer referring to 'The *Cassidulina teretis*  $\delta^{18}$ O ( $\delta^{18}$ Ob) confirms the relation between glacial stages and fine grained sediment as proposed by Kuhlman et al. (2006a,b)") Reply: Apart from our data, the benthic ( $\delta^{18}$ Ob) from the nearby Noordwijk well (Noorbergen et al., 2015) now independently confirms the relation between glacial stages and fine grained sediment as proposed by Kuhlman et al. (2006a). This has been added in the text.

L372 diverse. Reply: corrected

L377 Are herb and heath pollen dominant? Pinus remains the dominant species. Please make clear that you are discussing the pollen record, excluding pine pollen. Reply: comment added to highlight these are non-bisaccate forms only.

**L383 Which fresh water algae did you find?**

Reply: Pediastrum and Botryococcus (see supplementary data)

**L401-402 What does the n-C23 Sphagnum biomarker indicate?**

Reply: development of boreal (moist/cool) climate and influx, see also the reply to David Naafs.

**L413 MIS 96/95 (space missing)**

Reply: corrected

L422 Tables S and 2? Reply: corrected and updated

**Fig. 3 The Lingulodinium machaerophorum record should be presented separately – difficult to see now.**

Reply: we refrain from doing so as we do not want to expand the diagram any more. Also, the L. machaerophorum record is not critical to the interpretation.

**L428-: : : Chapter 6.1 is confusing and does not really deal with paleoenvironment. It is**

**not clear which MIS is discussed, and the switching between proxies (e.g. L429–433) and time intervals (all glacial/interglacials, MIS 98/97, 94 and 92) makes this difficult to follow.**

Reply: we have added discussion on the paleoenvironmental setting, revised the text for inconsistencies and changed the heading. We have also subdivided the chapter, see also the reply to the comment by reviewer 3.

**L432 depend (not depends)**

Reply; this suggestion is not correct. Pollen is a singulare tantum, it has no plural

**L434 Effect of SL on pollen is addressed here, but the effect of SL on the dinocyst record is not discussed in the MS**

Reply: the coastal dinocyst index is especially included to document the combined influence of coastal progradation and sea level change. Due to the earlier confusion on the use as warm water indicators, this point was perhaps overseen by the reviewer.

**L485, L535 Onset/intensification have been used intermixed**

Reply: valid point that we have adjusted to consistent use of intensification throughout the text

**L495–496 What is small – please specify? Please indicate which figure shows the small lead.**

Reply: In fig. 3, the lead between AP% decline and T/M increase is estimated between 3-8 kyr based on the present age model. This information has been added, see also comment to reviewer 3.

**L510 Speculation**

Reply: yes, but consistent with the effect of obliquity forcing

**518 Severe cooling. Subjective comment, certainly if you know that L. machaerophorum does not occur in regions with summer SST below 15°C. This species is present in all glacials**

Reply: the cooling is relative to late Pliocene conditions and in that respect severe, we do not exclude summer temperature above 15°C. In the conclusions we have specified more exactly the amplitude of cooling based on the brGDGT data that, although not always in phase with the other proxies, does give a temperature range for the G-IG cycles. Severe is changed to significant.

**Fig. S3 Please provide a list with the tiepoints:**

Reply: Table S1 has been added with all tiepoints based on Kuhlmann et al, 2006ab, uncertainties and references, together with an updated age depth model figure S3.

**Reviewer #2: David Naafs**

**General Comment (and Comment Line 82-88): Discussion on the phase relationship "the discussion on this specific topic in this manuscript is rather limited and is missing a discussion of crucial prior work on this topic"**

Reply : We originally aimed at providing a compact paper focusing on the evidence on phase relations we can provide from the new data, but we acknowledge that more information is available. We have expanded the introduction and discussion (updated section 6.3) on this matter following the suggestions of the reviewer to provide a more balanced assessment of the forcing mechanisms and available evidence. At the same time, we do not intend to provide a review paper and as such keep the discussion on additional literature limited.

**Comment: In addition, I wonder whether the age model is robust enough. The low-resolution benthic d18O record of this site does not always look like the LR04 stack.**

Reply: the validity of the age model is addressed in detail in the replies above to Stijn de Schepper, and we include more extensive discussion of records in the same basin, in particular the Noordwijk record from Noorbergen et al., 2015. The key results of our paper however, depend on the internal relations between the proxies from the same record and do not rely on an exact match with the LR04 stack.

Minor comments:

**Comment Line 51-55: this is a bit of a weird ending of the abstract, especially in the context of the main focus of the paper that is stated at the beginning of the abstract. The authors should end the abstract with a clear conclusion of what, according to their work, the phase relation is between forcing and climatic response.**

Reply: the abstract is adapted to better reflect the conclusion, but the last line is used to indicate that our observations have significance also for AMOC reconstructions, although not the topic of our study.

**Comment Line 66: a full review paper on IRD in the North Atlantic during the Plio/Pleistocene is given in (Naafs et al., 2013)**

Reply: reference has been added

**Comment Line 73-78: somewhere make reference to mechanism proposed**

Reply: reference added to proposed evaporation feedback forcing mechanism (Haug et al., 2005)

**Comment Line 202: what statistical basis was used to reject samples? What is the distinction between poor and not poorly preserved?**

Reply: see reply to comment of S. de schepper on this issue and additional methods in section 4.1

**Comment Line 280: cite (Eglinton and Hamilton, 1967) for odd over even predominance of nalkanes.** Reply: we have added Eglinton and Hamilton, 1967 on n-alkanes

**Comment Line 289 change sentence to "brGDGTs), produced by bacteria and that are abundant in soils, versus that:.."**

Reply: textual comments were adopted

**Comment Line 290: add reference**

Reply: we have cited Sinninghe Damsté et al., 2002 for crenarchaeol

**Comment Line 467: is there any other supporting information for the input of acidic peat input? For example, modern-day acidic peats are characterized by the dominance of the C31ab-hopane (Dehmer, 1995; Pancost et al., 2002), which is normally only present inmature sediments.**

Reply: as seen in the expanded pollen diagram (Fig. S2), Sphagnum spores are also mostly enhanced in the glacial MIS intervals in support of the C23 biomarker. We have analyzed part of the samples for the isomers index of the de C31 ab-hopanes. The results (for immature sediment) provide evidence of acidic peat input, although not dominant. Reviewer likely meant Pancost et al., 2003, which we have added

**Comment Line 473-477 The authors should provide a ternary plot of the brGDGT distribution to rule out a significant non-terrestrial contribution;**

Reply: a ternary brGDGT diagram has been made and added as Figure S6 in the supplement, and it is discussed in section 6.2 (new section heading).

**Comment Fig 3 readability**

Reply: the aim of the figure is to compare various proxies and they therefore need to be together. The figure is now rotated but will be horizontal in final version, improving visibility

**For the supplementary information, can the authors provide the abundances of the individual brGDGTs (and crenarchaeol) so that if the indices used for the soil-calibrations change in the future, the data can be easily recalculated and still be used in future studies.**

Reply: We have added the absolute abundances of the individual brGDGTs (and crenarchaeol) to enable recalculations in Table S3

**Referee #3: Anonymous**

Comment: The arboreal pollen and T/M ratio curve shows large fluctuations and hardly reveal any clear trends. These fluctuations may have resulted from a) the extremely low pollen sum after exclusion of bisaccate pollen, and b) the fact that the pollen results were merged from two different sites.

Reply: as Rev. #3 suggests, the AP curve shows variability. It is however clear that the glacial intervals AP values do not exceed 20%, except for one sample, and are consistently associated with increased Ericaceae. Interglacial AP values are clearly enhanced between 20 and 50 %, so we strongly disagree with the statement that there is no clear trend. The detailed pollen diagram in the supplementary data shows consistent abundance changes for the combined (spliced) dataset for e.g. Ericaceae, ferns, *Picea*. The variability in the *Pinus* curve is also visible in the sections that come from a single core, e.g. in MIS 95 and thus not a product of the splice. The splice is based on the high resolution GR record (verified by the dinocyst events), which provides a total of 15 tie points that produced a completely linear well tie (see Fig. 1). This figure has been added to the supplement as Fig. S4.

Figure 1: Well tie correlation points indicate a clear linear relation between the wells A15-3 and A15-4

**Comment: The enlarged figures show that many proxies were measured at different depths and with gaps, which at least for some intervals hamper a robust identification of leads and lags.**

Reply: we have indicated in figure S2 the key time lags between the palynological proxies that we use for the main interpretation of leading temperature change relative to sea level. The variable amount and resolution of samples is something we could not avoid as source material was limited and part was originally only produced for stratigraphic purposes, see also below the reply to comments 2/5.

Comment 1: An excellent age control is critical for all high-resolution studies of leads and lags. The authors should therefore provide more information on how the specific section has been dated. Reply: Our principal analysis of lead and lags are between proxies from the same record, and so essentially independent of age models, but in any case a solid age model is desirable. See the extensive reply on the age model to reviewer S. de Schepper, including tie points and age model construction. Section 3 has been expanded with additional information on the age model construction, and Table S2 with all chronostratigraphical tie points has been added.

Comments 3&4 The multiproxy approach makes the method chapter the longest section of the entire manuscript. Consider moving parts of the methods into the Supplementary Information and focus mainly on describing what the proxies show and discuss the methodological limitations relevant to this study. The palaeoenvironmental interpretation of the record lacks depth and should be more detailed.

Reply: we do provide the general basis of the interpretations for each proxy in the methods section of the original manuscript. We have expanded the palaeoenvironmental interpretation (and changed the heading), particularly on the pollen data, referring to the general depositional setting before discussing the climatological interpretation. We have retained the present manuscript organisation as it is likely the readers are not familiar with all proxies, and hence need to detailed descriptions.

Comments 2&5 It would be very helpful if the authors could provide a conceptual model describing in detail what they would expect to see in regard to the timing of each proxy, if obliquity forcing were the major driver. The analysis of lead and lags needs to be more detailed in order to provide convincing evidence for the main conclusion. I also struggle to see the parallel initial decrease of cold water dinocysts and Sphagnum biomarkers (first two curves) and the final decrease in T/M ratio and d180 (last two curves), which, according to the authors, followed with a delay of a few thousand years.

Reply: Statistical analysis of the lead-lag relations is desirable but unfortunately not possible due to the limits of the record recovery and very variable sample resolution between proxies due to limited source material, hence we choose to focus on the best resolved and completely cored G-IG cycle (MIS 98-97-96). In particular, as questioned by Rev#3; Decreases in cold water dinocysts and *Sphagnum* biomarkers (first two curves) and the final decrease in T/M ratio and d18O are based on high values of the first two in the early half of MIS 98 (shaded interval), after which the T/M increases only in the second half of MIS98 (and correlated LR04 d18O signal, but this detail depends on uncertainty in the age model). The key curves to assess are the cold water dinocysts and **not** the coastal signal, which is probably causing the

remarks by rev #3. Additional discussion on this topic in section 6.1 and indication of the main lead-lag relation between AP% and T/M ratio have been indicated in Fig. S2. The conceptual model has been expanded on in the introduction in combination with the more extensive literature discussion requested by reviewer Naafs, see for details the reply to that comment.

**Cited references (used in replies to all reviewers)**

Haug, G.H., Ganopolski, A., Sigman, D.M., Rosell-Melé, A., Swann, G.E.A., Tiedemann, R., Jaccard, S.L., et al., 2005. North Pacific seasonality and the glaciation of North America 2.7 million years ago. Nature 433, 821-825, doi: 10.1038/nature03332.

[revised manuscript text omitted]

- 22 The Netherlands
- 23 Correspondence to: t.h.donders@uu.nl
- 24

25

**26 Abstract**

| 27   | We assess t

---

## Author Response (AR2)

**Reply to editor corrections (R2)**

Comment  Figure 1: your study site is indicated with a red asterisk. It isn't easily distinguished on the figure in its current size, and there is a danger that the asterisk will be very difficult to identify if the figure is reduced in size during proofing. Is it possible for you to make this symbol larger / more bold / outlined in black so that it isn't lost in the graphic?
**Reply**: we have changed the figure and inserted a larger asterisk outlined in black in Figure 1

Comment - line 515: write "excluding" in full rather than "excl"
**Reply**: corrections made

Comment - Figure 3: can you clarify in the caption what the red line is on the T/M ratio? I'm assuming some smoothing function but i couldn't find an explanation in the text or supplemental figures.
**Reply**: explanation added in caption fig3, line is a LOWESS smoother

Comment - lines 581-583: can you clarify the time frame of this delta front advance, since it is not clear from this text nor from the figure caption (are you talking about patterns of change restricted to the same time interval covered by your record, or something longer term?)
**Reply**: we added the age range for the main advance in the main text (discussion) and in the figure caption of Supl. Figure 1

Comment - lines 584-585: sentence starting "In- or exclusion..". I would recommend trying to edit this sentence as it is a bit confusing. Perhaps "Bisaccate pollen is the component most sensitive to differential transport processes, yet regardless of whether it is included in the T/M index (Fig. S5) the same patterns are recorded, indicating no direct influence of differential transport on the T/M ratio" ?
**Reply**: we adopted the suggestion of the editor

Comment - throughout the manuscript there are some incidents of "G-IC cycle" being used, but I assume this should be "G-IG cycle" if you are discussing glacial-interglacial cycles?
**Reply:** Use of G-IC corrected

Comment - supplemental figure S3: can you clarify in the caption which of these tie points is from which method e.g. are those with the large age uncertainties the biostratigraphy tie points? Were all tie points used (Table S1 seems to suggest some big discrepancies between e.g. dinocyst datums and the others which are listed). Different colours / shading for different methods might help on the graphic.
**Reply**: we have adapted supl Figure 2 indicating the type of age tie point. The figure does not contain the Miocene ages from Table S1 as they are prior to a hiatus, which has now been explained in the caption. The only other point omitted form the figure is described as very scattered in the table, and hence not useable.

Comment - Table S1 (chronostratigraphic tie points): can you clarify in the caption whether the first 'age'

column is the age that you have assigned to that particular depth in A15-3? (and then the min/max age as being those assigned to the datum in the literature?).

**Reply**: we have changed the header and information in the table caption, range indicates indeed the min and max from recent literature.

[revised manuscript text omitted]

**Supplementary figures and data descriptions**

**Supplementary Figure 1:** sedimentary facies interpretation of high-resolution seismically
mapped surfaces S4-6 (see Fig. 2 for stratigraphical position) in relation to the wells.
Approximate ages of the mapped surfaces are 2.8 Ma (S4), 2.44 Ma (S5) and 2.34 Ma (S6).

[Figure]

**Supplementary Figure 2**: Main pollen types in the spliced record of A15-3 (blue) and A15-4
(red) expressed as percentages of the total pollen. Dashed lines between AP % and T/M ratio
indicate the observed lags of 3-8 kyr between (terrestrial) cooling and sea level decreases.

[Figure]

**Supplementary Figure 3:**

Age-depth model of the spliced A15-3 and A15-4 sections based on a smoothing spline
interpolation (optimised range set to 1.65) of tie age points in Table S1 taken from Kuhlmann
et al. (2006ab). The tie points where updated to recent range calibrations where necessary.

[Figure]

**Supplementary Figure 4:**

Well tie correlation points between A15-3 and A15-4 are based on the Gamma Ray
correlation displayed in Fig. 2. The high $R^2$ of the linear relation between the well tie points
confirms that the proxy records from both wells can by spliced confidently into a single
record.

[Figure]

**Supplementary Figure 5:**

The terrestrial to marine palynomorph (T/M) ratio with in- and exclusion of bisaccate pollen
show minimal offset, indicating a low influence of differential transport processes.

[Figure]

**Supplementary Figure 6:**

Ternary diagram based on the brGDGT analyses showing values close but not identical to the soil calibration. Samples with higher BIT (>0.5) show a greater correspondence to the soil calibration, which indicates some contribution of in-situ produced (aquatic) brGDGTs.

[Figure]

**Supplementary Table S1**

Chronostratigraphical control points taken from Kuhlmann et al. (2006ab). The tie points where updated to recent range calibrations where necessary. Minimum and maximum ages represent the range of ages from literature. Plotted ages represent the midpoints of these values. Miocene ages are not shown in Fig. S3 due to a hiatus at ~1240 mbsl separating the Miocene and Pliocene deposits (Kuhlmann et al., 2006ab).

**Supplementary Table S2**

Palynological abundance data of A15-3 and A15-4 by taxon and composite indices as displayed in Fig. 3. Percent data are based on a separate marine and terrestrial total.

**Supplementary Table S3**

Borehole Gamma Ray data (A15-3) and geochemical data of A15-3 and A15-4 (O and C stable isotope ratios, TOC, Alkane ratios, brGDGTs).